# CTCF loss has limited effects on global genome architecture in *Drosophila* despite critical regulatory functions

Anjali Kaushal [1,14], Giriram Mohana [1,14], Julien Dorier [2,14], Isa Özdemir [1], Arina Omer[3], Pascal Cousin[1], Anastasiia Semenova[1], Michael Taschner [4], Oleksandr Dergai[1], Flavia Marzetta[2,5], Christian Iseli [2], Yossi Eliaz[3,6,7], David Weisz [3], Muhammad Saad Shamim [3,8,9], Nicolas Guex [2], Erez Lieberman Aiden[3,7,10,11,12,13 ✉] & Maria Cristina Gambetta [1 ✉]

Vertebrate genomes are partitioned into contact domains defined by enhanced internal contact frequency and formed by two principal mechanisms: compartmentalization of transcriptionally active and inactive domains, and stalling of chromosomal loop-extruding cohesin by CTCF bound at domain boundaries. While *Drosophila* has widespread contact domains and CTCF, it is currently unclear whether CTCF-dependent domains exist in flies. We genetically ablate *CTCF* in *Drosophila* and examine impacts on genome folding and transcriptional regulation in the central nervous system. We find that CTCF is required to form a small fraction of all domain boundaries, while critically controlling expression patterns of certain genes and supporting nervous system function. We also find that CTCF recruits the pervasive boundary-associated factor Cp190 to CTCF-occupied boundaries and co-regulates a subset of genes near boundaries together with Cp190. These results highlight a profound difference in CTCF-requirement for genome folding in flies and vertebrates, in which a large fraction of boundaries are CTCF-dependent and suggest that CTCF has played mutable roles in genome architecture and direct gene expression control during metazoan evolution.

[1] Center for Integrative Genomics, University of Lausanne, Lausanne, Switzerland. [2] Bioinformatics Competence Center, University of Lausanne, Lausanne, Switzerland. [3] The Center for Genome Architecture, Baylor College of Medicine, Houston, TX, USA. [4] Department of Fundamental Microbiology, University of Lausanne, Lausanne, Switzerland. [5] Vital-IT, SIB Swiss Institute of Bioinformatics, Lausanne, Switzerland. [6] Department of Physics, University of Houston, Houston, TX, USA. [7] Center for Theoretical Biological Physics, Rice University, Houston, TX, USA. [8] Department of Bioengineering, Rice University, Houston, TX, USA. [9] Medical Scientist Training Program, Baylor College of Medicine, Houston, TX, USA. [10] Department of Molecular and Human Genetics, Baylor College of Medicine, Houston, TX, USA. [11] Departments of Computer Science and Computational and Applied Mathematics, Rice University, Houston, TX, USA. [12] Broad Institute of MIT and Harvard, Cambridge, USA. [13] Shanghai Institute for Advanced Immunochemical Studies, Shanghai Tech University, Shanghai, China. [14] These authors contributed equally: Anjali Kaushal, Giriram Mohana, Julien Dorier. ✉email: erez@erez.com; mariacristina.gambetta@unil.ch

A wide range of animal genomes are partitioned into a series of contact domains (CDs) that exhibit increased physical proximity among loci within them. An evolutionarily conserved mechanism of such genome folding is thought to be compartmentalization, reflecting the segregation of chromosomal domains based on their transcriptional and epigenetic states[1–3]. In vertebrates, chromosomal loops are additionally extruded on underlying compartment domains through a process involving DNA-bound CTCF molecules which stall loop-extruding cohesin complexes at domain boundaries[1,4–10]. CTCF-dependent extrusion-based boundaries either reinforce or counteract compartmental domain boundaries, depending on the locus. Overall, a large fraction of boundaries in the vertebrate genome are CTCF-dependent[9,11].

Intriguingly, although *Drosophila* has widespread CDs and CTCF, it is currently unclear whether CTCF-dependent domains exist in *Drosophila*. High-resolution genome-wide Hi-C maps of formaldehyde-crosslinking frequencies between pairs of DNA fragments (as a measurement of their proximity in 3D-space) were recently generated in *Drosophila* tissue culture cells[2,12–15]. These studies highlighted the lack of hallmarks of CTCF-mediated domains observed in vertebrate cells. Rather, evidence suggests that CDs in flies are formed by CTCF-independent compartmentalization and other transcription-related processes, as most boundaries lie between domains with different histone modifications or at promoters of highly transcribed genes[2,12,16–18].

Crucially, the functional importance of genome folding into CTCF-dependent domains is not fully understood in any organism. CTCF is essential for the viability of mammalian cells[11,19,20], whereas it is dispensable for early development in *Drosophila*[21]. Assessing whether or not CTCF-mediated domains exist in *Drosophila* is important for understanding their relevance for genome function. Recent studies have perturbed specific CDs in flies to address their biological roles without knowing whether they are CTCF-mediated or compartmental[22–24], yet different types of CDs may have different functions.

CTCF-dependent domains in mammals generally comprise regulatory elements and their target promoters[25–27]. This suggested that CTCF somehow limits regulatory crosstalk between CDs, and fosters regulatory interactions within them. This model is, however, difficult to test in mammals because global perturbation of CTCF leads to cell death. Acute depletion of CTCF protein in mouse embryonic stem cells followed by transcriptional profiling did not reveal widespread transcriptional changes[11]. Alternatively, deletion of CTCF binding sites near developmental genes in cultured cells and mice identified some sites where CTCF appears to critically prevent developmental defects and disease[28–30], and many CTCF sites that did not appear functional[31–33]. These diverse results paint an opaque picture of how CTCF impacts gene expression. Previous studies that partially knocked-down CTCF in *Drosophila* cell lines also did not reveal clear effects on transcription[34–36]. Analysis of the homeotic phenotype of CTCF[0] mutants completely lacking both maternal and zygotic CTCF suggested that CTCF blocks regulatory crosstalk between elements on either side of some CTCF binding sites[21]. A fundamental question arising from comparative studies in flies and humans is how CTCF impacts transcription, and how this relates to its uncertain architectural function in flies. Whether CTCF stably associates with partner proteins to effect its functions also remains unclear.

Here, we show using CTCF[0] mutant *Drosophila* that CTCF is critically required in neurons for fly viability. We examine the effects of CTCF loss on genome folding and transcriptional regulation in the central nervous system (CNS) and investigate the molecular basis of CTCF function.

## Results

### CTCF expression in neural stem cells (NSCs) or neurons is essential for fly viability.

To identify a biologically relevant tissue in which to study CTCF function in *Drosophila*, we used previously described CTCF knock-out (CTCF[KO]) mutants and CTCF[0] mutants that additionally lack maternally inherited CTCF[21]. Some CTCF[KO] mutants (60%) hatch into adults with spasmatic movements suggesting a neurological phenotype that might be the cause of their short lifespan (Figs. 1a, 1b, Supplementary Movie 1). We tested the relevance of CTCF expression in the nervous system by performing tissue-specific knock-out and rescue experiments. Specifically, we used Gal4 drivers active in NSCs, mature neurons or muscles to drive conditional excision of a CTCF rescue transgene (knock-out) or UAS-CTCF expression (rescue) in CTCF mutant genetic backgrounds. Loss of CTCF expression in NSCs or neurons compromised the ability of flies to hatch to a comparable extent as loss of all zygotic CTCF expression (Fig. 1a) and severely shortened the life span of flies that did hatch (Fig. 1b, Supplementary Movie 2). On the other hand, loss of CTCF in muscle only slightly impaired adult hatching and life span (Figs. 1a, b).

In contrast to CTCF[KO], CTCF[0] mutants never hatch from the pupal case (Fig. 1c). Conditional expression of CTCF in NSCs or neurons of CTCF[0] mutants strongly rescued hatching (Fig. 1c) and adults were capable of coordinated movements and survived for several days (Fig. 1d, Supplementary Movie 3). On the other hand, expressing CTCF in muscles of CTCF[0] mutants barely rescued hatching (Fig. 1c, d).

Together, these results show that CTCF expression is critically required in neurons for pupal hatching and adult viability. Consistently, CTCF is more highly expressed in the nervous system than in other tissues[37,38]. Analyses of molecular phenotypes of CTCF[0] mutants described hereafter were therefore performed in dissected CNSs of third instar larvae, a developmental stage at which CTCF[0] mutants are fully viable.

### Physical insulation defects in CTCF[0] mutants.

To address whether CTCF is required to form CD boundaries in flies, Hi-C was performed on CNSs dissected from wildtype (WT) and CTCF[0] larvae in biological triplicate using two 4-cutter restriction enzymes for enhanced resolution. Hi-C maps consisting of 200 million reads per genotype were obtained by combining the correlated biological replicates (see Methods, Supplementary Table 1). Hi-C maps from whole bodies of single flies of the same genotypes were also generated. In parallel, CTCF binding sites were mapped in larval CNSs by chromatin immunoprecipitation sequencing (ChIP-seq) with a polyclonal antibody specifically recognizing CTCF (Supplementary Fig. 2a) in WT and in CTCF[0] animals as control. Only 740 CTCF peaks were defined as enriched in WT relative to CTCF[0] CNSs, of which 77% overlapped a CTCF consensus motif (Supplementary Fig. 2b, Supplementary Data 1).

To assess the relation between CTCF peaks and CD boundaries genome-wide in WT CNS Hi-C maps, boundaries were identified at 2 kb resolution with TopDom (see "Methods", Supplementary Table 2, Supplementary Data 2 and 3). Very few (<1%) boundaries defined in this study potentially correspond to small CDs defined in even higher resolution Hi-C studies (see "Methods"). Domain boundaries were enriched within ±1 kb of several (36%) CTCF peaks (Fig. 2a). Conversely, a CTCF peak was located within ±1 kb of only 8% of all boundaries (Fig. 2b). This indicates that while CTCF peaks are frequently at domain boundaries, CTCF is only present at a small fraction of all boundaries in flies.

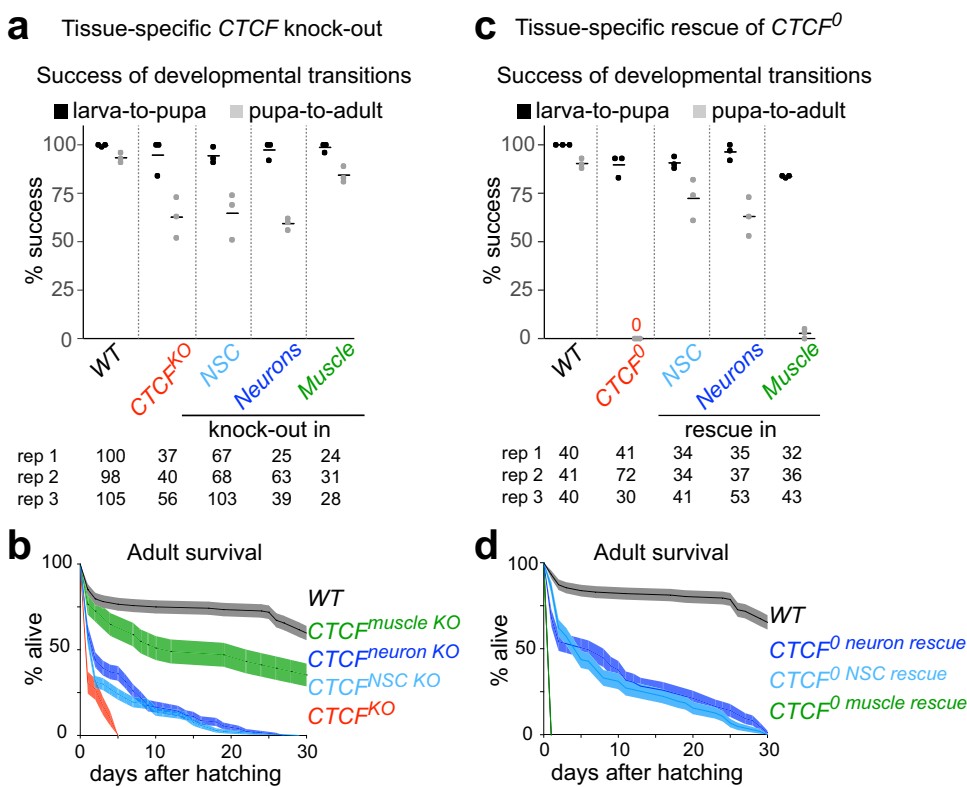

**Fig. 1 CTCF expression in neural stem cells or neurons is essential for *Drosophila* viability. a** Percentages (in y) of wildtype (WT), $CTCF^{KO}$ and animals lacking CTCF in neural stem cells (NSCs), neurons or muscle that successfully transition from third instar larva to pupa (black) and from pupa to adult (gray) in $n = 3$ biological replicates (rep 1–3), each containing the indicated number of animals per genotype. Horizontal lines show means. (**b**) Percentage of live adults of the same genotypes up to 30 days after pupal hatching. 40 animals of each genotype were analyzed in triplicate; dark lines show mean and shading shows ±standard deviation. **c** Same as (**a**) but for WT, $CTCF^0$ and animals with CTCF expression restricted to NSCs, neurons or muscle. **d** Same as (**b**) but for genotypes indicated in (**c**).

WT and $CTCF^0$ Hi-C maps were globally similar, and most (84%) domain boundaries were detected in both WT and $CTCF^0$ mutants. Nevertheless, specific CDs were visibly less physically insulated from the neighboring domain in $CTCF^0$ mutants (Fig. 2c, Supplementary Fig. 2c, Supplementary Table 3). Clearly disrupted domain boundaries in $CTCF^0$ mutants frequently occurred at former CTCF peaks (Fig. 2d). Of 135 strongly affected domain boundaries that were lost in $CTCF^0$ mutants, 89 (66%) were at former CTCF peaks (Supplementary Table 2). To determine how generally physical insulation defects are observed at former CTCF peaks in the absence of CTCF (irrespective of their localization at CD boundaries identified by TopDom), physical insulation score differences between WT and $CTCF^0$ mutants were measured across all 740 CTCF peaks. Boundary defects in $CTCF^0$ mutants were observed at most former CTCF peaks, with more prominent defects visible at CTCF peaks that are highly occupied in WT (Fig. 2e, Supplementary Fig. 2d). CTCF-dependent boundaries were variably positioned relative to neighboring genes (see examples in Fig. 2c: CTCF peaks 2, 3, 5 and 6 are respectively in an intron, at the end of a gene, within 1 kb of a gene promoter or intergenic). Many CTCF-dependent boundaries were similarly affected in Hi-C maps from whole-body flies of the same genotypes, indicating that CTCF is required to form physical boundaries in most cell types (Supplementary Fig. 2e). Together, these results strongly suggest that CTCF mediates the formation of physical boundaries.

Whereas domain boundaries were abolished at several former CTCF peaks in $CTCF^0$ mutants, they were partially retained at other peaks that are similarly occupied by CTCF in WT (Supplementary Fig. 2c, compare boundary defects at CTCF peaks 5 and 6). Of 343 WT CD boundaries bound by CTCF, only 125 (36%) were fully lost in $CTCF^0$ mutants (Supplementary Table 2). This resulted in a lower average physical insulation score at former CTCF peaks in $CTCF^0$ mutant CNS Hi-C maps (Fig. 2f). These observations are not due to the presence of contaminating CTCF, as CTCF RNA and protein are undetectable by RNA-seq and ChIP-seq (Fig. 2c and next section). As $CTCF^0$ mutants lack CTCF from the beginning of development, residual boundaries can also not be explained by a role of CTCF in the establishment but not maintenance of boundaries. Rather, this observation suggests that at some sites, CTCF reinforces boundaries redundantly established by other mechanisms, a scenario also observed in mammalian cells[1,2]. We define CTCF-occupied CD boundaries present only in WT as strictly CTCF-dependent, and those that are present in $CTCF^0$ (generally weaker than in WT) as partially CTCF-dependent. These two types of CTCF-dependent boundaries are contrasted later in the "Results" section.

A region in the N-terminus of human CTCF directly interacts with cohesin and stabilizes cohesin on DNA[10,39], partly explaining how human CTCF forms CD boundaries. Vertebrate and fly CTCF N-termini are highly diverged, yet a 10 amino acid residue stretch in CTCF's N-terminus that binds to cohesin in human cells is present at a similar distance from the zinc finger domain in fly CTCF[10] (boxed in Supplementary Fig. 2f). We therefore tested whether two residues critical for cohesin interaction in human CTCF (Y226 F228, homologous to Y248 F250 in fly CTCF) mediate direct interaction of fly CTCF with the SA-Vtd (homologous to human SA2-SCC1) complex. For this, GFP-tagged recombinant WT and Y248A F250A point mutant

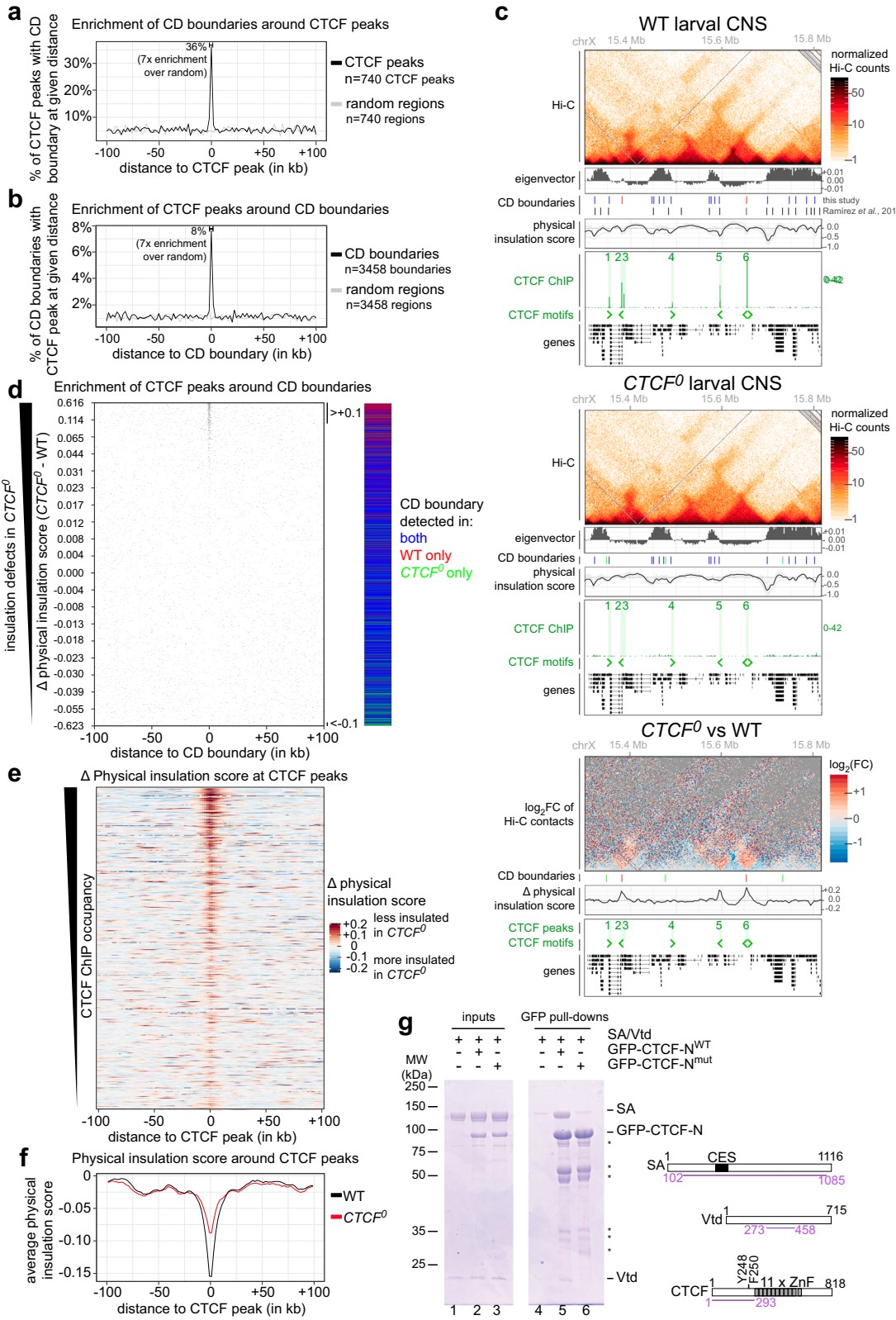

CTCF N-termini were mixed with an untagged SA-Vtd subcomplex and purified on GFP binder beads. WT but not mutant CTCF versions retained SA-Vtd (Fig. 2g). Therefore, despite profound divergence, the fly CTCF N-terminus interacts directly with cohesin in vitro. This interaction was suggested to impart directionality to CTCF-dependent boundaries in mammalian cells[10,39], but we find that CTCF has at best a very weak

preference to establish directional boundaries (Supplementary Fig. 2g) consistent with a previous study[2].

We conclude that *Drosophila* CTCF is required to form physical boundaries with strengths generally proportional to its occupancy on DNA. Other mechanisms reinforce CTCF-dependent boundaries at some sites and explain the formation of most boundaries in flies.

**Fig. 2 Physical insulation defects in *CTCF⁰* mutants. a** Percentage of $n = 740$ CTCF peaks with at least one contact domain (CD) boundary at a given distance (per 2 kb bins) around the CTCF peak. Enrichment of CD boundaries around the same number of random positions (gray) is shown as control. **b** Percentage of $n = 3458$ CD boundaries with at least one CTCF peak at a given distance (per 2 kb bins) around the CD boundary. Enrichment of CTCF peaks around the same number of random positions (gray) is shown as a control. **c** Example locus (dm6 coordinates) Hi-C maps, eigenvector values (positive for A compartment, negative for B compartment), CD boundaries from this study (color-coded as in Fig. 2d) and a Hi-C study in cultured cells[17], physical insulation score (calculated with different window sizes in gray, average in black), CTCF ChIP-seq (CTCF peaks highlighted and numbered), CTCF motif orientations in DNA, and gene tracks in WT (top) and *CTCF⁰* (middle) larval CNSs. (Below) Differential (*CTCF⁰* minus WT) Hi-C map and physical insulation score. **d** Position of CTCF peaks around all CD boundaries defined in any genotype ($n = 3970$ boundaries) ranked by physical insulation score differences measured in *CTCF⁰* minus WT Hi-C maps. Visibly weaker boundaries in *CTCF⁰* (score > +0.1) or in WT (score < −0.1) are bracketed. Boundaries are color-coded in all figures as present in both WT and *CTCF⁰* (blue), only in WT (red) or only in *CTCF⁰* (green). **e** Physical insulation score differences measured in *CTCF⁰* minus WT Hi-C maps around CTCF peaks, ranked by CTCF ChIP occupancy in WT. **f** Average physical insulation scores around CTCF peaks in WT (black) and *CTCF⁰* (red). **g** GFP pull-down of tagged CTCF N-terminus (residues 1–123) that is WT (GFP-CTCF-N^WT) or Y248A F250A point mutant (GFP-CTCF-N^mut) mixed with untagged recombinant cohesin subcomplex (residues 102–1085 of SA and 273–458 of Vtd). Specific retention of cohesin by CTCF (lane 5) is higher than the background binding of SA-Vtd to beads (lanes 4, 6). Asterisks mark GFP-CTCF-N degradation. CES conserved essential surface, ZnF zinc finger.

---

**CTCF impacts expression patterns of genes near CTCF peaks.** To understand how CTCF impacts transcription, we performed RNA sequencing (RNA-seq) on cDNA libraries from mRNA purified from WT and *CTCF⁰* larval CNSs in triplicate. This confirmed the absence of *CTCF* mRNA in *CTCF⁰* samples (Supplementary Fig. 3a). 392 (~3% of all) genes were significantly differentially expressed (DE) in *CTCF⁰* mutants (with adjusted p-value<0.05 and |fold-change| > 1.5) (Fig. 3a, Supplementary Data 4). *CTCF⁰* mutants therefore do not show widespread transcriptional defects, though changes occurring in subsets of cells in the CNS such as CTCF's previously validated target gene *Abdominal-B* elude our analysis[21].

Some DE genes had decreased expression in *CTCF⁰* mutant CNSs compared to WT (Fig. 3b). Several DE genes with increased expression in *CTCF⁰* CNSs are normally not expressed in the CNS but rather restricted to other specialized tissues like testes (*Intraflagellar transport 52*), tendons (*Thrombospondin*), and the peripheral nervous system (*Odorant receptor 67d*) (Figs. 3c, 3d, Supplementary Fig. 3b). Some ectopic transcripts lacked annotated start and termination sites suggesting that they are cryptic (Supplementary Fig. 3b). RNA fluorescent in situ hybridization (RNA-FISH) analysis showed that genes with increased expression in *CTCF⁰* CNSs were misexpressed in various patterns, possibly driven by locus-specific enhancers (Fig. 3e).

Indirect transcriptional changes are expected in *CTCF⁰* mutants, which lack CTCF since the beginning of development, and we asked whether CTCF regulates genes in the vicinity of its binding sites. 10% of DE genes had a CTCF peak within ±1 kb of their transcriptional start site (TSS) (ninefold enrichment over randomly sampled matched non-DE genes) (Fig. 3f), a result that was not very different for genes with increased versus decreased expression in *CTCF⁰* mutants (Supplementary Fig. 3c). Conversely, 5% of CTCF peaks were located within ±1 kb of a DE gene TSS (9-fold enrichment over randomly sampled matched non-DE genes) (Fig. 3g). These results suggest that, depending on the locus, CTCF may directly repress or activate the transcription of nearby genes, or alternatively CTCF may shield promoters from inappropriate enhancers or silencers as observed at *Hox* gene loci[21,40].

Could the structural defects observed in *CTCF⁰* Hi-C maps be secondary consequences of gene misregulation in the vicinity of former CTCF peaks? Some CTCF-dependent domain boundaries were located far from genes (Fig. 2c, CTCF peak 6 is 9 kb away from the closest gene) and are thus unlikely to be impacted by transcription. Others were located near genes whose expression increased (Supplementary Fig. 2c, peak 3), decreased (Supplementary Fig. 2c, peak 6) and in most cases remained unchanged (Supplementary Fig. 2c, peak 7). Few (8%) DE genes were located

in different A/B compartments in *CTCF⁰* mutants relative to WT, indicating that differential gene expression mostly occurred without large changes in higher-order spatial chromatin configuration (Supplementary Fig. 3d, Supplementary Data 5). Together, these results indicate that the pervasive weakening of physical boundaries observed at former CTCF peaks in *CTCF⁰* mutants (Fig. 2e-f) is not a mere consequence of altered transcription.

**CTCF occupancy scales with enhancer-blocker activity in a reporter assay.** Previous studies of the functionality of CTCF binding sites stably integrated into the fly genome suggested that most of them lack insulator activity (i.e., the ability to block regulatory crosstalk)[36], at least in single copies[40]. Here, we tested CTCF peaks in a quantitative reporter assay. The reporter comprises an enhancer positioned between two fluorescent reporter genes (EGFP and mCherry) driven by minimal *Heat-shock-protein-70* (*Hsp70*) promoters (Fig. 4a). Test fragments were cloned in between EGFP and the enhancer, maintaining the enhancer at a similar distance from both reporter genes. Reporter plasmids were then transiently transfected into *Drosophila* S2 cells, and relative EGFP and mCherry intensities were measured in thousands of single cells with a cell analyzer (Supplementary Fig. 4a). An insulator should reduce EGFP expression while mCherry expression should remain high. Control experiments with a neutral spacer or the well-characterized *gypsy* insulator[41] validated the assay (Fig. 4b, lanes 1 and 2). Two CTCF peaks near genes whose expression decreased (peak G from Fig. 3b) or increased (peak N from Fig. 3e) in *CTCF⁰* mutants had similar effects as *gypsy* (Fig. 4b, lanes 3 and 4). EGFP levels in the presence of *gypsy* or CTCF peaks were not strongly reduced below basal levels measured in enhancer-less control reporters (Supplementary Fig. 4b), suggesting that these tested sequences mostly impaired enhancer-mediated EGFP expression. Additional CTCF peak regions (Supplementary Fig. 4c, average size 360 bp) were tested and their relative insulator strengths were estimated from the median ratio of mCherry-over-EGFP fluorescence measured in single cells. Eleven out of 14 tested CTCF peaks selectively reduced EGFP intensities to various degrees that globally scaled with CTCF ChIP-seq occupancy measured in S2 cells[42] (Fig. 4c) and that appeared independent of the endogenous locations of CTCF peaks relative to their nearest genes (Supplementary Fig. 4c) and of combinatorial co-binding with other fly insulator-binding proteins on the cloned fragments (Supplementary Fig. 4d). Mutating two base pairs of a CTCF motif in one of these fragments abolished its activity (Fig. 4c, fragment N mut); thus, the reporter specifically reveals the activity of a single CTCF

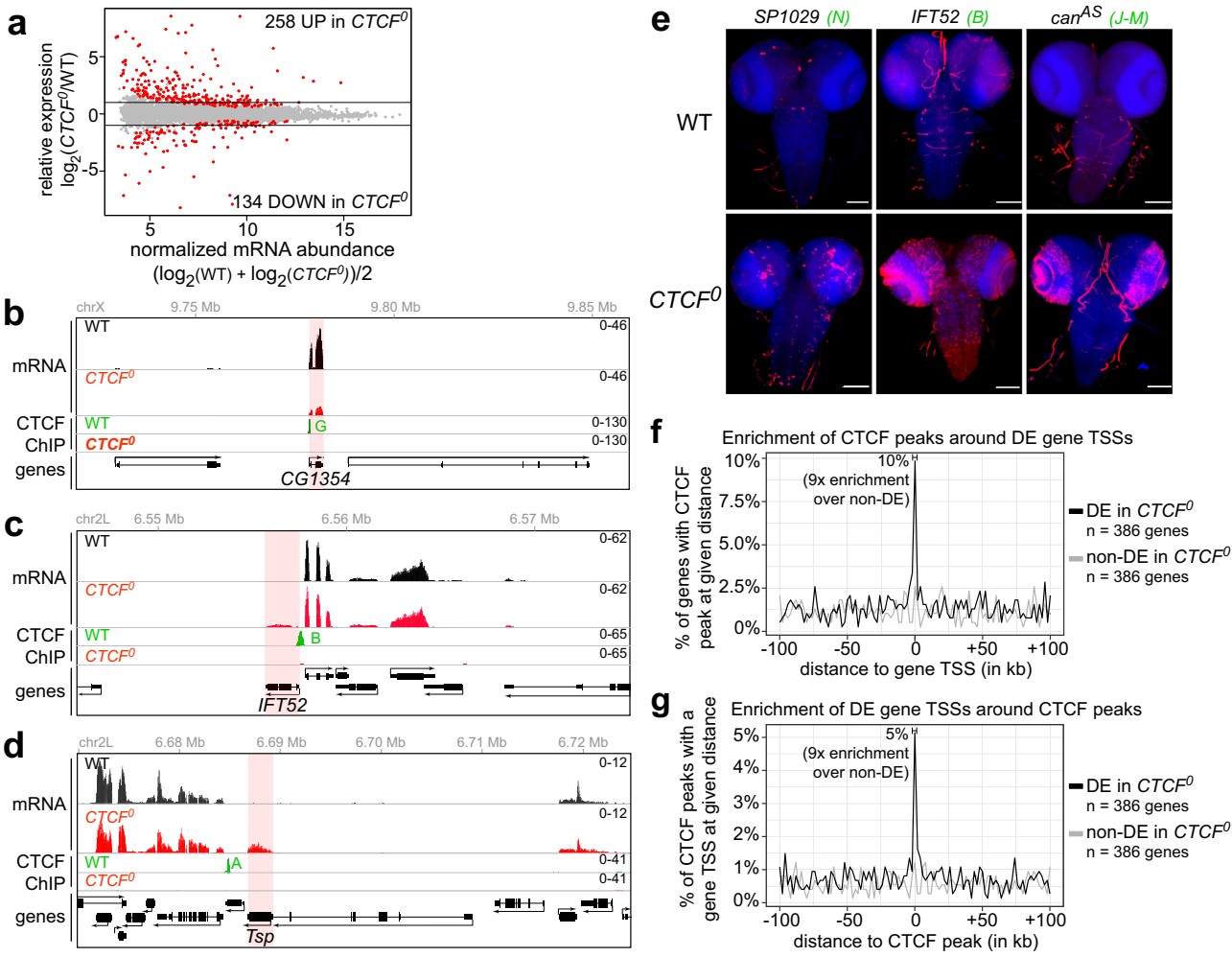

**Fig. 3 CTCF impacts expression patterns of genes near CTCF peaks. a** RNA-seq MA plot of $CTCF^0$ versus WT CNSs with mean abundance (in x) plotted as a function of enrichment (in y). Differentially expressed (DE) genes ($p$adj < 0.05 and |fold change| > 1.5) are red. **b–d** RNA-seq signals in WT (black) and $CTCF^0$ (red) larval CNSs, and CTCF ChIP-seq signals in WT (green) and $CTCF^0$ (red) larval CNSs at CG1354 (**b**), IFT52 (**c**) and Tsp (**d**) loci. Differentially transcribed regions are shaded in red. Scales in tracks of all figures indicate reads per million. In all figures, CTCF peaks labeled by capital letters were tested in Fig. 4c. **e** RNA-FISH with antisense probes (red) against indicated transcripts in CNSs of wildtype and $CTCF^0$ larvae stained by DAPI (blue) (scale bars 100 μm). mRNAs of SP1029, IFT52 and an antisense transcript overlapping can (shown in Supplementary Fig. 3b) are normally not expressed in wildtype CNSs (background signal is sometimes visible in trachea) and are misexpressed in different patterns in $CTCF^0$ mutants. All animals showed similar misexpression patterns for a given transcript. **f** Percentage (in y) of $n = 386$ DE genes in $CTCF^0$ larval CNSs (black) or $n = 386$ randomly sampled expression-level-matched non-DE genes (gray) with at least one of 740 CTCF peaks at a given distance (per 2 kb bins) around the gene TSS, measured in the direction of transcription (in x). Ten percent of DE genes have at least one CTCF peak within ±1 kb of their TSS, which is ninefold higher than the average enrichment at the sampled non-DE genes. **g** Percentage (in y) of CTCF peaks with at least one of $n = 386$ DE gene TSSs (black) or $n = 386$ randomly sampled expression-level-matched non-DE gene TSSs (gray) at a given distance (per 2 kb bins) around CTCF peaks, measured in the direction of transcription (in x). Five percent of CTCF peaks have at least one DE gene TSS within ±1 kb, which is 9-fold higher than at the sampled non-DE TSSs.

binding site. Taken together, these observations indicate that CTCF sites in the reporter do not strongly directly repress or activate transcription but rather insulate a promoter from an enhancer.

**CTCF recruits Cp190 to a subset of Cp190-bound domain boundaries.** To further understand how CTCF functions, we asked whether it stably associates with partner proteins that contribute to its activity. Unbiased identification of CTCF partners from *Drosophila* embryonic nuclear extracts in biological duplicates by mass spectrometry reproducibly identified known insulator-binding proteins Centrosomal protein 190 kDa (Cp190) and Insulator binding factors 1 and 2 (Ibf1 and Ibf2) as enriched CTCF interactors relative to negative control (Supplementary

Fig. 5a). Reciprocal Cp190 purifications published by others also identified Ibf1, Ibf2 and CTCF among other proteins[43]. Traces of the cohesin complex also co-purified with CTCF (Supplementary Fig. 5a) reminiscent of transient interactions between cohesin and CTCF seen in mammalian cells[44].

CTCF was previously shown to directly interact with Cp190[45], yet the relevance of this interaction remained unclear. No common target genes are known[46] and a mutant version of CTCF reported to no longer interact with Cp190 was largely functional *in vivo*[45]. We performed pull-downs of GFP-tagged CTCF fragments co-expressed in bacteria with Cp190's BTB (Broad-Complex, Tramtrack and Bric-a-brac) domain and found that amino acids 698-771 in CTCF C-terminus directly interact with Cp190 BTB (Supplementary Fig. 5b). Importantly, this stretch in

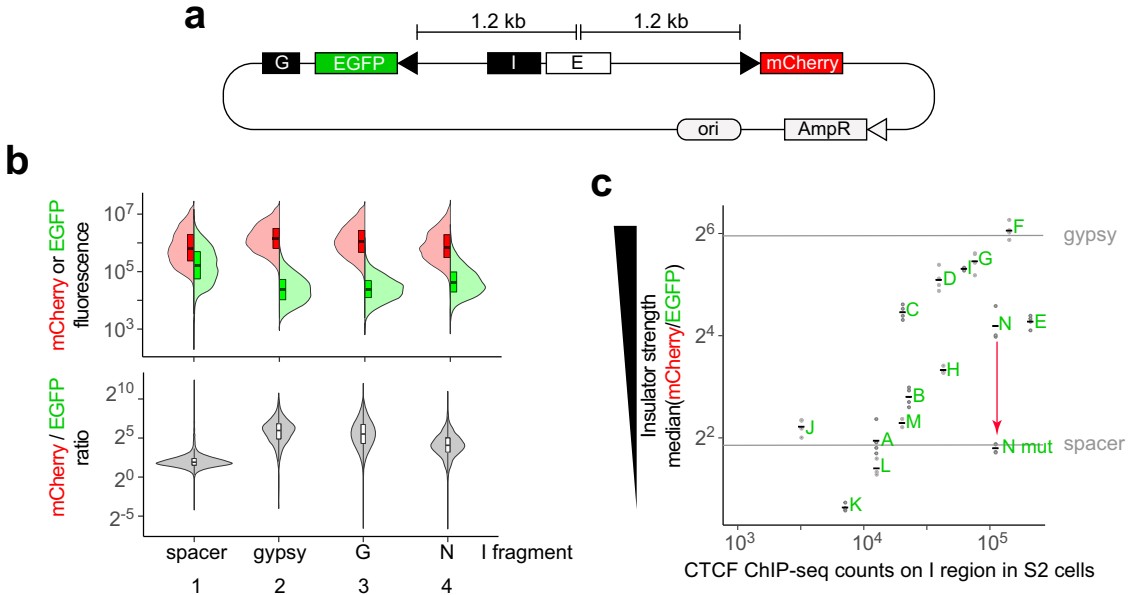

**Fig. 4 CTCF occupancy scales with enhancer-blocker activity in a reporter assay. a** In the reporter plasmid, a test insulator I is cloned in between an enhancer E and EGFP, and mCherry serves as a reference (elements are drawn to scale, arrowheads represent *Hsp70* minimal promoters). A *gypsy* insulator G is present downstream of EGFP to block EGFP activation by the enhancer (which in a circular plasmid molecule is both upstream and downstream of EGFP) from the left. **b** Split violin plots (thick lines mark medians, boxes mark interquartile ranges) show distributions of mCherry (left) and EGFP (right) fluorescence intensities (log₁₀ values in y) measured in thousands of single S2 cells transiently transfected with reporters with indicated I fragments (in x). mCherry-to-EGFP ratios (log₂ values in y) in single cells are shown below. For each reporter, merged biological triplicates are plotted. **c** Median mCherry-to-EGFP ratios in single transfected S2 cells (log₂ values in y) relative to CTCF ChIP-seq counts in S2 cells [42] (log₁₀ values in x) on selected CTCF peaks (labeled A–N, Supplementary Fig. 4c) cloned as I fragments. n = 2 (M), 3 (H, I, L, N) or 4 (A–G, J, K, N mut) biological replicates (dots) and mean values (horizontal lines) are shown. As a reference, mean values obtained with the *gypsy* insulator or a neutral spacer are indicated as horizontal lines. As a control, a CTCF motif in fragment N was mutated, leading to fragment N mut for which CTCF ChIP occupancy was not determined.

CTCF does not overlap the previously deleted region (amino acid residues 774–818) that was used to conclude that CTCF's interaction with Cp190 was unimportant in vivo.

To assess the genome-wide overlap between CTCF and Cp190 binding sites in larval CNSs, specific Cp190 peaks were identified by ChIP-seq with a polyclonal anti-Cp190 antibody in WT and in *Cp190^KO* animals with a CRISPR-Cas9 mediated deletion of the *Cp190* open reading frame as control (Supplementary Fig. 5c). 6,473 Cp190 peaks were enriched in WT relative to *Cp190^KO* CNSs (Fig. 5a, Supplementary Data 6). Cp190 colocalized with CTCF at most (79%) CTCF peaks and was additionally present at many other sites (Fig. 5a), consistent with other studies[35,36,47]. We profiled Cp190 binding sites in WT and *CTCF^0* larval CNSs and found that Cp190 was normally recruited to most Cp190 peaks in *CTCF^0* mutants with the exception of former CTCF peaks, at which Cp190 was globally reduced (Figs. 5a, 5b, Supplementary Data 7 and 8). In *CTCF^0* mutants, Cp190 was lost from former higher-occupancy CTCF peaks but only reduced at former lower-occupancy CTCF peaks (Fig. 5b). We therefore distinguish between strictly CTCF-dependent Cp190 peaks (lacking a detectable Cp190 peak when comparing *CTCF^0* and *Cp190^KO* mutants) and partially CTCF-dependent Cp190 peaks (with a detectable Cp190 peak in *CTCF^0* relative to *Cp190^KO* mutants, generally weaker in *CTCF^0* than in WT).

Unlike CTCF, Cp190 binding was enriched at CD boundaries genome-wide (Fig. 5c lane 3, Supplementary Figs. 5d, e)[2,15,17]. Outside of CTCF peaks, Cp190-occupied domain boundaries were often proximal to transcribed TSSs (Fig. 5c, lane 6). In *CTCF^0* mutants, residual Cp190 binding at former CTCF-occupied boundaries was significantly associated with boundary retention (Figs. 5d–f). Seventy-five percent of strictly CTCF-dependent boundaries lacked a residual Cp190 peak, and 80% of

residual Cp190 peaks were associated with a residual boundary in *CTCF^0* mutants (Fig. 5e). CD boundary defects in *CTCF^0* mutants were also less severe at former TSS-proximal CTCF peaks (within 200 bp of a gene TSS) than at former TSS-distal CTCF peaks (Fig. 5f). This suggests that either Cp190 itself, its associated factors, or transcription at Cp190-bound TSSs may redundantly contribute to the formation of physical boundaries independently of CTCF and may synergize with CTCF at partially CTCF-dependent Cp190 peaks (see examples in Fig. 5g).

**CTCF and Cp190 co-regulate a subset of target genes**. To assess whether loss of Cp190 results in transcriptional changes shared with *CTCF^0* mutants, RNA-seq was performed on *Cp190^KO* larval CNSs in biological triplicate. Overall, 440 DE genes were observed in *Cp190^KO* mutant CNSs compared to WT, of which 192 went up and 248 went down relative to WT (with adjusted *p*-value < 0.05 and |fold-change| > 1.5) (Supplementary Fig. 6a, Supplementary Data 9). Since Cp190 is bound to many more sites than CTCF (Fig. 5a), we did not expect that many transcriptional changes in *Cp190^KO* mutants would be shared in *CTCF^0* mutants. Surprisingly, however, a considerable fraction of DE genes in *CTCF^0* and *Cp190^KO* mutants were common (31% of all DE genes in *CTCF^0* and 26% of all DE genes in Cp190^KO) and concordantly changed in similar directions and to similar degrees relative to WT (Fig. 6a). This is exemplified at the *SP1029* (Fig. 6b–c) and *CG15478* (Fig. 6d–e) genes that are proximal to a CTCF and Cp190 co-bound peak (peak 1/N in Fig. 6b, peak 2 in Fig. 6d). In the absence of CTCF, Cp190 is additionally lost from these peaks (Figs. 6b and d, middle), a CD boundary is disrupted (Supplementary Figs. 6b and c), and the gene is expressed at increased (*SP1029* in Fig. 6b, middle) or decreased (*CG15478* in Fig. 6d, middle) levels relative to

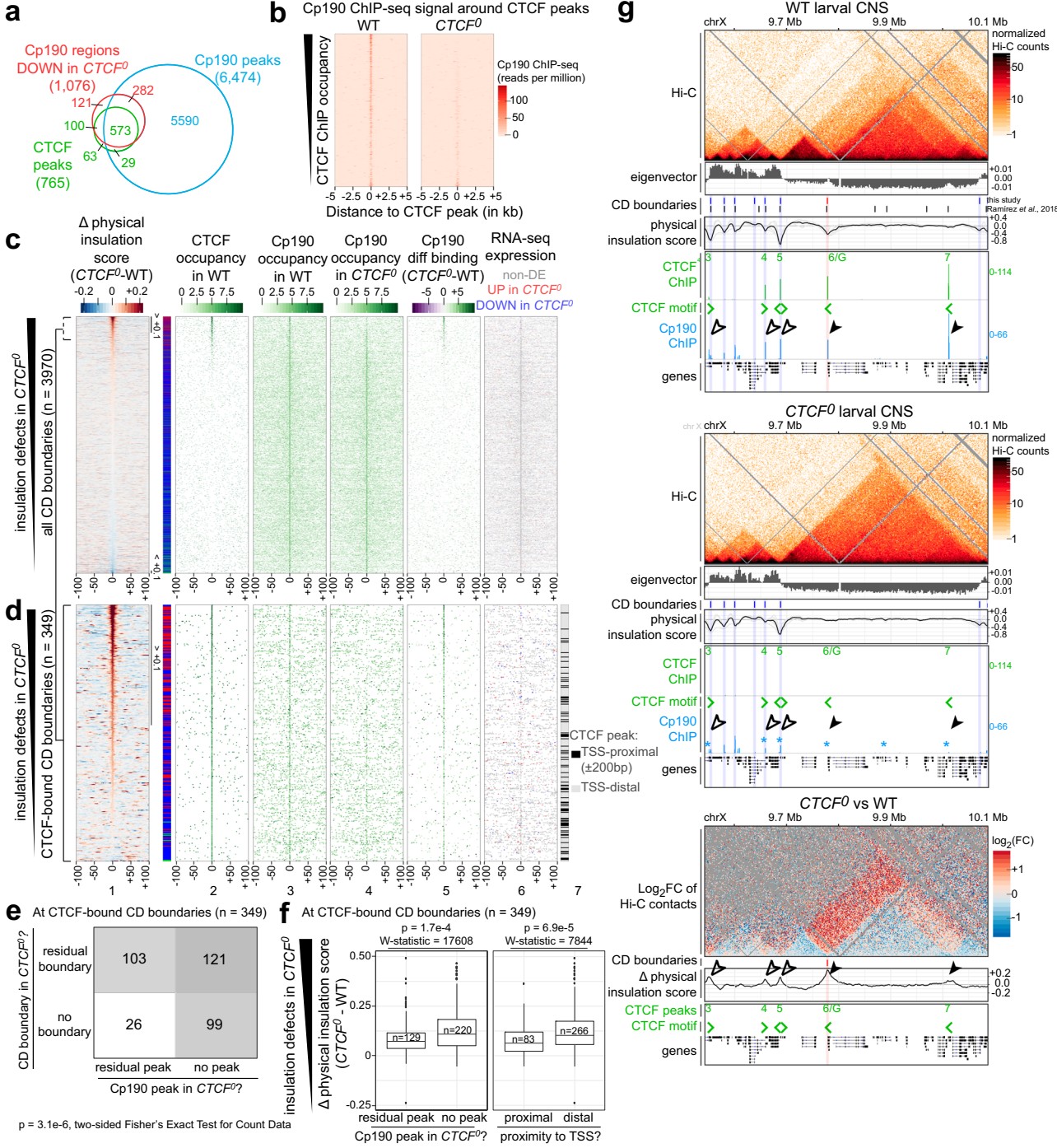

WT. In the absence of Cp190, CTCF remains bound at *SP1029* (Fig. 6b, bottom) and *CG15478* (Fig. 6d, bottom) which are nevertheless also similarly misexpressed relative to WT (Figs. 6b and d, bottom). This suggests that Cp190 is required for CTCF function independently of CTCF binding to DNA. To more stringently compare *SP1029* and *CG15478* misexpression in the absence of CTCF or Cp190, we visualized their mRNAs in embryos completely lacking maternal and zygotic CTCF (*CTCF⁰*) or Cp190 (*Cp190⁰*). Already at 11 h of development, *CTCF⁰* and *Cp190⁰* embryos ectopically expressed *SP1029* in the same cells (in the nervous system and additional cell types) (Fig. 6c) and failed to express WT levels of *CG15478* in the nervous system (Fig. 6e). We conclude that Cp190 is a critical partner of CTCF for regulating a subset of common genes (see summary model in Fig. 6f).

## Discussion

CTCF-dependent CDs have been proposed to regulate the communication between genes and their regulatory elements. Here, we analyzed *Drosophila* that developed in the complete absence of CTCF and reached the following conclusions: (1) CTCF is most critically required in neuronal cells for adult viability (Fig. 1). (2) Domain boundary defects in *CTCF⁰* mutants are overwhelmingly associated with CTCF-bound sites, consistent with a mechanism in which CTCF can form boundaries (Fig. 2). At the same time, the vast majority of boundaries are CTCF-independent. (3) CTCF prevents ectopic activation and silencing of certain genes in its vicinity (Fig. 3). (4) Sites bound by CTCF do not directly repress or activate transcription, but rather functionally insulate promoters and enhancers in a reporter assay in S2 cells (Fig. 4). (5)

**Fig. 5 CTCF recruits Cp190 to a subset of Cp190-bound domain boundaries. a** Overlap between CTCF (green) and Cp190 (blue) peaks in WT, and regions with reduced Cp190 binding in $CTCF^0$ relative to WT (red). Some peaks were split for three-way comparisons (see "Methods"). **b** Cp190 ChIP-seq signal in WT or $CTCF^0$ around CTCF peaks, ranked by CTCF occupancy in WT. **c** Distribution of indicated datasets around CD boundaries defined in any genotype ($n = 3970$ boundaries) ranked by insulation defects in $CTCF^0$. (1) Insulation score differences in $CTCF^0$ minus WT. Visibly weaker boundaries in $CTCF^0$ (score > +0.1) or in WT (score < −0.1) are bracketed. On the right, boundaries are classified as in Fig. 2d. (2–4) ChIP occupancy of CTCF peaks in WT, Cp190 peaks in WT or Cp190 peaks in $CTCF^0$. (5) Differential Cp190 ChIP occupancy in $CTCF^0$ minus WT. (6) Expressed TSSs in WT and $CTCF^0$ with similar (gray), increased (red) or decreased (blue) expression in $CTCF^0$ relative to WT. **d** As above for CD boundaries with a CTCF peak within ±2 kb ($n = 349$ boundaries) centered on the closest CTCF peak classified as TSS-proximal (within ±200 bp of a TSS) or distal (lane 7). **e** Numbers of CD boundaries bound by CTCF in WT ($n = 349$ boundaries) that are present or absent in $CTCF^0$ mutants, and whose associated CTCF peak overlaps or not a residual Cp190 peak in $CTCF^0$ mutants (p-val = 3.1e−6, two-sided Fisher's Exact Test for Count Data). **f** Physical insulation score differences in $CTCF^0$ minus WT at CTCF-bound CD boundaries ($n = 349$ boundaries) are higher when the associated CTCF peak does not overlap a residual Cp190 peak in $CTCF^0$ mutants, or a TSS within 200 bp (indicated p values and W-statistics from two-sided Wilcoxon rank-sum test with continuity correction). Box plot: center line, median; box limits, upper and lower quartiles; whiskers, 1.5× interquartile ranges; points, outliers; $n$ = CTCF peaks of each category (in x). **g** Example locus like Fig. 2c also displaying Cp190 ChIP-seq signal in WT and $CTCF^0$ mutant larval CNSs. Asterisks mark Cp190 peaks in $CTCF^0$ mutants with reduced occupancy relative to WT revealed by differential analysis. Solid arrowheads mark strictly CTCF-dependent boundaries (the second boundary was not called by TopDom), empty arrowheads mark partially CTCF-dependent boundaries.

Cp190 directly binds to the C-terminus of CTCF and is recruited to CTCF peaks in a strictly or partially CTCF-dependent manner (Fig. 5). Residual Cp190 binding at former CTCF peaks coincides with residual boundary retention in $CTCF^0$ mutants (Fig. 5). (6) CTCF binding to DNA alone is not sufficient for correct expression patterns of a subset of genes that also rely on Cp190. Below we discuss how this work furthers our understanding of genome folding in *Drosophila*, CTCF's role in transcriptional regulation and the molecular basis thereof.

**Relaxed requirement of CTCF for *Drosophila* genome architecture.** In comparison to vertebrates, the principles of genome folding into CDs in *Drosophila* are less clear. On the one hand, the majority of fly CDs were proposed to form by compartmentalization of domains with different transcriptional states or because actively transcribed genes cluster, with little contribution from architectural proteins acting independently of transcription[2,48]. On the other hand, analyses of enriched transcription factor motifs at domain boundaries defined at high-resolution revealed that 77% were enriched in core promoter motifs (and called promoter boundaries) and the remaining 23% were enriched in motifs of insulator-binding proteins like CTCF, su(Hw) and Ibf1 (and called non-promoter boundaries)[17]. This suggested that architectural proteins may form some domain boundaries. By completely ablating CTCF in vivo, we definitively show that CTCF contributes to the formation of a small fraction (below 10%) of domain boundaries in *Drosophila* (Fig. 2). This strongly contrasts with the mammalian genome where extrusion-based mechanisms are responsible for the formation of a large fraction of boundaries. This demonstrates that although domain formation is ubiquitous in different species, the contributions of different mechanisms can vary widely. The limited role that CTCF plays in global genome architecture in flies is nevertheless consistent with our finding that CTCF binding sites are an order of magnitude less frequent in flies (~800 peaks in 130 Mb genome) than in humans (~80,000 peaks in 3 billion bp genome)[49], and the fact that alternative boundary-forming mechanisms exist in flies.

At strictly CTCF-dependent boundaries, CTCF can form boundaries independently of the presence/absence of a nearby TSS and of detectable transcriptional changes in nearby genes (Figs. 2c and 5d). At partially CTCF-dependent boundaries, defects in $CTCF^0$ mutants are limited by redundant boundary-forming mechanisms often associated with CTCF-independent recruitment of Cp190, Cp190-associated factors or the presence of Cp190-bound transcribed gene TSSs (Figs. 5c–g and 6f). Cp190 marks both promoter and non-promoter boundaries (Fig. 5c)[15,17],

and it remains to be clarified whether Cp190 or its associated factors directly contribute to domain boundary formation (through similar or unrelated mechanisms as CTCF) or whether boundary formation is governed by transcription of Cp190-bound TSSs. Pervasive transcriptional perturbation globally affects Hi-C contact maps[2,16,48], indicating that transcription itself or the transcription machinery at least reinforces CDs. Finally, we note that apart from CTCF, the transcription factor Zelda has also been shown to affect CD boundaries in flies: Zelda depletion in early *Drosophila* embryos led to partial disruption of former Zelda-occupied domain boundaries, and to concurrent loss of RNA polymerase II recruitment which may account for the observed boundary defects[16].

Whether *Drosophila* CTCF, like its mammalian counterpart, forms CD boundaries in concert with loop-extruding cohesin remains unclear because of discrepancies between flies and mammals. (1) In mammalian Hi-C maps, CTCF sites at both anchors of an extruded loop often engage in high-frequency contacts[4] not seen in *Drosophila*[2] (Fig. 2c, Supplementary Fig. 2c). (2) CTCF and cohesin colocalize genome-wide in mammals[49,50], but cohesin does not colocalize specifically with CTCF in *Drosophila*[13,17]. Fly CTCF may therefore not have a robust or unique ability to stall or stabilize loop-extruding cohesin complexes, despite their ability to interact in vitro (Fig. 2g). (3) CTCF-dependent boundaries are directional in mammals[4,5,51] but lack clear directionality in flies (Supplementary Fig. 2g)[2]. All these discrepancies could nevertheless be expected given the probable differences in how fly CTCF interacts with extruding cohesin (Supplementary Fig. 2f). Indeed, previous in silico simulations[6] and experiments affecting loop-extrusion processivity across CTCF-dependent boundaries in human cells[7,9,10] described CDs with weaker corner interactions more similar to domains observed in flies. The N-terminus of DNA-bound mammalian CTCF may stall or stabilize cohesin by directly interacting with cohesin subunits and regulators[10,39,52,53] via binding interfaces that are not all conserved in fly orthologs (Supplementary Fig. 2f). Our results suggest that direct interaction of fly CTCF N-terminus with cohesin is insufficient to form directional chromosomal loops.

**Impact of CTCF on transcriptional regulation.** Functional studies of how CTCF impacts expression are challenging in mammalian cells. Recent studies that manipulated CTCF binding sites at specific loci have moderated our view of how critical CTCF is for patterned gene expression, but a limitation is that effects can be masked by unperturbed CTCF sites nearby that function redundantly[31–33].

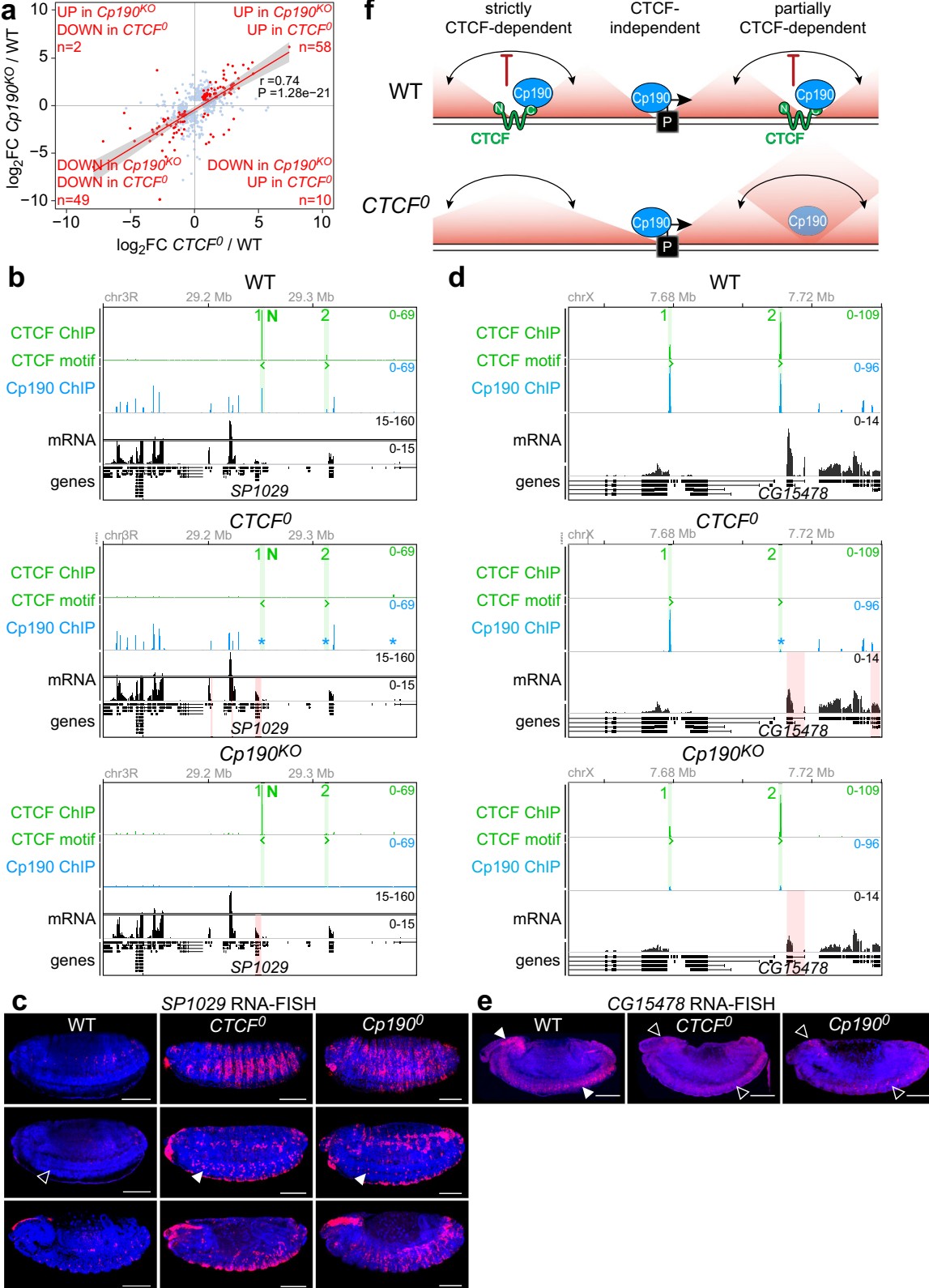

Our transcriptional analyses of *Drosophila CTCF⁰* CNSs showed that CTCF is required for patterned expression of selected genes in the CNS while at the same time being dispensable for orchestrating other complex gene expression programs. Gene misexpression may result from defective gene insulation from local regulatory elements, as supported by the binding of CTCF between certain neuronal and non-neuronal genes in vivo (Figs. 3c, d), the increased expression of these genes in *CTCF⁰* larval CNSs (Figs. 3c–e) and the enhancer-blocking activity of CTCF peaks in S2 cells (Fig. 4b–c). Our reporter assay is independent of chromatin environment, allowing quantitative measurements of insulator activity that reveal a direct relation to

**Fig. 6 CTCF and Cp190 co-regulate a subset of target genes. a** DE genes (with padj<0.05 and |fold change| > 1.5) in *CTCF^O* and/or *Cp190^KO* mutant larval CNSs relative to WT with detectable expression in both differential RNA-seq analyses (omitting 55 DE genes in *CTCF^O* and 54 DE genes in *Cp190^KO* that had low counts in the other differential analysis) are plotted in light blue and red. DE genes common in *CTCF^O* and *Cp190^KO* mutants are highlighted in red and counted in each quadrant. Pearson's correlation coefficient and *p*-value show correlated changes of common DE genes. The red line shows linear regression and gray shadowing the corresponding 95% confidence interval. **b** Extended *SP1029* gene locus displaying CTCF ChIP-seq (peaks numbered and highlighted in green), CTCF motif orientations in DNA, Cp190 ChIP-seq, and mRNA-seq tracks (DE genes highlighted in red) in WT (top), *CTCF^O* (middle) and *Cp190^KO* CNSs (bottom). Asterisks mark Cp190 peaks in *CTCF^O* mutants with reduced occupancy relative to WT revealed by differential analysis. **c** Lateral views of 11 h old embryos of labeled genotypes (columns) in 3 confocal sections (rows) subjected to *SP1029* RNA-FISH (scale bars 100 μm). Arrowheads mark *SP1029* misexpression in the nerve chord of *CTCF^O* and *Cp190^O* mutants (filled arrowheads), not occurring in WT embryos (empty arrowhead). **d** As Fig. 6b for the extended *CG15478* gene locus. Residual Cp190 ChIP signal in *Cp190^KO* mutants could be maternally deposited Cp190 or non-specific ChIP signal. **e** As Fig. 6c for *CG15478* RNA-FISH. Arrowheads mark *CG15478* expression in the brain and nerve chord of WT embryos (filled arrowheads), strongly reduced in *CTCF^O* and *Cp190^O* mutants (empty arrowheads). **f** Wildtype *Drosophila* contact domain boundaries are strictly CTCF-dependent, partially CTCF-dependent, or not bound by CTCF. CTCF recruits Cp190 to CTCF-dependent boundaries, and Cp190 is recruited independently to additional boundaries many of which are close to transcribed gene promoters. In *CTCF^O* mutants, Cp190 is lost from strictly CTCF-dependent boundaries, while at other former CTCF peaks residual Cp190 binding is associated with partial boundary retention. CTCF-dependent boundaries can prevent regulatory crosstalk (double-sided arrows) between genes and regulatory elements positioned on either side, and Cp190 co-regulates a subset of genes together with CTCF.

the efficiency of CTCF recruitment. These findings are consistent with our previous characterization of *Hox* gene misexpression in *CTCF^O* mutants, which phenocopies deletions of insulator boundaries that maintain the independence of some *Hox* regulatory domains[21]. Our ability to detect gene misregulation in *CTCF^O* larval CNSs likely depends on genomic context, notably the presence of regulatory elements active in this organ in a sufficiently large number of cells to detectably alter transcription.

Why aren't gene misexpression defects in *CTCF^O* mutants more widespread? Recent studies have emphasized that specific communication between regulatory elements and gene promoters is controlled at many levels, of which CTCF provides one. In particular, enhancer-promoter compatibility[54] and regulation of the chromatin properties of regulatory elements themselves[55] also determine whether or not regulatory elements and promoters functionally communicate. CTCF may also function redundantly with other insulator-binding proteins in *Drosophila* to limit regulatory crosstalk in this compact genome. Unlike what is known in mammals, flies have a family of insulator-binding proteins, many of which have DNA binding domains with which they target specific loci[56].

**Molecular basis of how CTCF impacts gene regulation**. Whether CTCF's ability to form physical boundaries explains its conserved genetic insulator activity remains an open question[1,57]. An ideal scenario to address this would be to separate boundary formation from gene insulator function. Human CTCF with mutated critical cohesin-interacting residues was largely functional, but CD boundaries were only partially disrupted[10]. We observed that some DE genes in *CTCF^O* mutants are close to partially CTCF-dependent boundaries (Fig. 5d, lane 6). Gene misregulation in the absence of CTCF may therefore occur despite significant retention of a physical boundary, but we did not definitively confirm that these DE genes are direct CTCF targets.

We found that CTCF functionally cooperates with a stably bound regulatory cofactor, expanding the view of how CTCF may impact gene regulation. The relevance of the CTCF-Cp190 interaction has been debated. On the one hand, Cp190 was assumed to be required for CTCF's insulator function based on the observations (1) that the enhancer-blocking activity of a *Hox* gene insulator in transgenic reporter assays depended on both CTCF and Cp190, and (2) that CTCF failed to be recruited to many sites on polytene chromosomes in *Cp190* mutants[58,59]. The latter observation was, however, not reproduced in genome-wide ChIP experiments in Cp190 knock-down cells[36]. On the other

hand, no common CTCF and Cp190 target genes were known[46], and the interaction between CTCF and Cp190 was recently concluded to be dispensable in vivo[45]. The latter conclusion was based on deleting residues in CTCF that did not interact with Cp190 in our pull-down experiments (Supplementary Fig. 5b). We identified genes with concordant transcriptional changes upon loss of either CTCF and Cp190 that are potentially directly regulated by both proteins.

Is this interaction conserved in vertebrates? Around 40 Cp190-like proteins comprising an N-terminal BTB domain and zinc fingers exist in humans[60], but Cp190 does not have a direct ortholog. The C-terminus of human CTCF is capable of interacting with the BTB domain of a Cp190-like protein called KAISO in yeast two-hybrid experiments[61], reminiscent of the interaction between fly CTCF C-terminus and the BTB domain of Cp190 (Supplementary Fig. 5b). Whether CTCF transiently interacts with a BTB domain-containing protein in human cells or whether this interaction has not been maintained in vivo remains to be clarified.

How do Cp190 and CTCF collaborate? Incomplete overlap of DE genes in *CTCF^O* and *Cp190^KO* mutants suggests that CTCF requires Cp190 at some loci but not others (Fig. 6a). Alternatively, additional common targets may be masked by other transcriptional changes in *Cp190^KO* mutants or by maternal Cp190 rescuing early defects in these mutants. How Cp190 functions is not known, but it may contribute to CTCF's insulator activity similarly to how Cp190 contributes to the activities of *gypsy* and some *Hox* gene boundary insulators[46,62]. Cp190 may help CTCF form CD boundaries, or Cp190 may function independently of boundary formation through unknown mechanisms that could uncover paradigms for controlling the communication between genes and regulatory elements.

## Methods
**Tissue-specific *CTCF* loss-of-function**. (*CTCF^KO*, *UAS-FLP*)/*TM6B* heterozygotes were crossed to *CTCF^KO*/*TM6B* heterozygotes for an independently isolated *CTCF^KO* allele that also carried an FRT-flanked genomic *CTCF* rescue transgene and one of various Gal4 drivers: expressed in neuroblasts [*worniu-Gal4* (Bloomington stock 56553)], mature neurons [*elav-Gal4* (Bloomington stock 25750)], or muscles [*Mef2-Gal4* (Bloomington stock 25756)]. Resulting non-TM6B animals were transheterozygous for *CTCF^KO* alleles, derived from a WT maternal germline, and expressed *UAS-FLP* under the control of a Gal4 driver leading to tissue-specific excision of the *CTCF* rescue transgene. *w^1118* (wildtype) and *CTCF^KO* transheterozygous animals were used as controls.

**Tissue-specific rescue of *CTCF^O* mutants**. Females trans-heterozygous for two independently isolated *CTCF^KO* alleles were rescued with an FRT-flanked genomic *CTCF* rescue transgene that was excised in their germline by expressing FLP

recombinase under the control of *nanos* regulatory sequences. These females were crossed to *CTCF^{KO}*/TM6B males carrying a *UAS-CTCF-3xHA* transgene (FlyORF stock F000619) and a Gal4 driver mentioned above or no Gal4 driver as control. Resulting non-TM6B animals were transheterozygous for *CTCF^{KO}* alleles, derived from a maternal germline devoid of *CTCF* (*CTCF^0* mutant background) and expressed *UAS-CTCF* under the control of a Gal4 driver. $w^{1118}$ animals were used as WT control.

**Drosophila viability tests.** Three sets of 30–40 third instar larvae of desired genotypes were transferred into separate vials and the number of pupae and fully hatched adults was recorded. The average percentage and standard deviation of animals alive at each developmental stage and over a 30-day period after hatching were scored and plotted in Kaplan-Meier survival plots with 5% confidence intervals from the triplicate experiments.

**Antibodies.** For this study, polyclonal rabbit antibodies were raised against CTCF[1–293] and Cp190[1–1096]. Proteins were recombinantly purified in *E. coli* by tandem affinity purification using N-terminal GFP- and C-terminal His-tags. Tags were cleaved off by 3C protease and used for immunization.

**Western blotting.** Forty third-instar larval CNSs per biological replicate were dissected in ice-cold PBS. Samples were sonicated in 100 μl of 20 mM Tris pH 7.5, 500 mM NaCl, 0.1% Triton X-100, 1× complete protease inhibitors (Roche) in a Bioruptor (settings on high, 5 min, 4 °C). Extracts were centrifuged for 5 min at maximum speed and total protein was quantified by Qubit protein assay (ThermoFisher). Calibrated amounts of extract from WT, *CTCF^0* and *CTCF^{OE}* animals were loaded on a 4–12% acrylamide gel and probed with rabbit anti- CTCF[1–293] crude serum (diluted 1:1000) and mouse anti-tubulin clone B-5-1-2 (Sigma T5168, diluted 1:10,000). *CTCF^{OE}* animals expressed a CTCF cDNA under the control of upstream activating sequences (UAS) driven by a ubiquitous *tubulin-Gal4* driver, and served as control. Chemilumiscence pictures of nitrocellulose membranes were imaged in Fiji v2.1.0/1.53c.

**Chromatin preparation from larval CNSs.** 60 third-instar larval cuticles per biological replicate (two biological replicates per sample except CTCF ChIP-seq in WT performed in biological triplicates) were dissected in ice-cold PBS, then cross-linked 15 min at room temperature in 1.8% (v/v) paraformaldehyde, 50 mM HEPES pH 8, 100 mM NaCl, 1 mM EDTA, 1 mM EGTA. Crosslinking was stopped by washing for 10 min in 1 ml PBS, 0.01% Triton-X100, 125 mM glycine, then cuticles were washed for 10 min in 10 mM HEPES pH 7.6, 10 mM EDTA, 0.5 mM EGTA, 0.25% Triton X-100. CNSs were dissected from the cuticles in 10 mM HEPES pH 7.6, 200 mM NaCl, 1 mM EDTA, 0.5 mM EGTA, 0.01% Triton X-100, then sonicated in 120 μl of RIPA buffer (10 mM Tris-HCl pH 8, 140 mM NaCl, 1 mM EDTA, 1% Triton X-100, 0.1% SDS, 0.1% sodium deoxycholate, protease inhibitor cocktail) in AFA microtubes in a Covaris S220 sonicator for 5 min with a peak incident power of 140 W, a duty cycle of 5% and 200 cycles per burst. Sonicated chromatin was centrifuged to pellet insoluble material and snap-frozen.

**ChIP-seq.** ChIP was performed with 2 μl of rabbit polyclonal antibody crude sera against CTCF[1–293] or Cp190[1–1096], each incubated with half of the chromatin prepared from a biological replicate overnight at 4 °C. Twenty-five microliters of pre-mixed Protein A and G Dynabeads (Thermo Fisher 100-01D and 100-03D) were added for 3 h at 4 °C, then washed for 10 min each once with RIPA, four times with RIPA with 500 mM NaCl, once in LiCl buffer (10 mM Tris-HCl pH 8, 250 mM LiCl, 1 mM EDTA, 0.5% Igepal CA-630, 0.5% sodium deoxycholate) and twice in TE buffer (10 mM Tris-HCl pH 8, 1 mM EDTA). DNA was purified by RNase digestion, proteinase K digestion, reversal of crosslinks at 65 °C for 6 h, and elution from a QIAGEN Minelute PCR purification column. ChIP-seq libraries were prepared using the NEBNext Ultra II DNA Library Prep kit for Illumina. An equimolar pool of multiplexed ChIP-seq libraries at 4 nM was sequenced on the Illumina HiSeq4000 (150 bp paired-end).

**ChIP-seq analysis.** Paired-end ChIP-seq reads were demultiplexed and mapped to the dm6 genome using Micmap, a derivative of the fetchGWI tool[63]. Only chromosomes 2, 3, 4, and X were used. ChIP-seq peaks were called using the R package csaw[64] v1.16.1 using a window width of 20 bp and spacing of 10 bp, ignoring duplicate reads. A background enrichment was evaluated as the median over all samples in the comparison of the average number of reads per 2 kb bins. Windows with less than threefold enrichment over background were filtered out. Data were normalized using the TMM method[65] implemented in csaw. Differential binding analysis in csaw is based on the quasi-likelihood framework implemented in the edgeR package[66]. Results obtained on different windows were combined into regions by clustering adjacent windows. Combined *p*-values were evaluated for each region using csaw and Benjamini & Hochberg method was applied to control the false discovery rate. Regions with false discovery rate (FDR) < 0.01 and |fold change| > 2 were considered as differential binding regions and are reported in Supplementary Data files 1, 6, 7, and 8. Genuine CTCF peaks were identified by differential analysis of ChIP-seq signals in WT versus *CTCF^0* as being lower in the

mutant samples relative to WT. Genuine Cp190 peaks were similarly identified by differential analysis of ChIP-seq signals in WT versus *Cp190^{KO}* (Cp190 peaks in WT) or in *CTCF^0* versus *Cp190^{KO}* (Cp190 peaks in *CTCF^0*). Additional differential analyses were performed for Cp190 ChIP-seq signal in WT versus *CTCF^0* (for Fig. 5a). We defined ChIP occupancy as the best.log2FC obtained from csaw in the respective differential analysis. We defined peak positions as the best.pos obtained from csaw. To count overlaps between CTCF and Cp190 peaks in three-way comparisons shown in Fig. 5a, some CTCF and Cp190 peaks were split into 2 or 3 sub-regions. Specifically, 740 WT CTCF peaks were split into 765 peaks, 6473 WT Cp190 peaks were split into 6474 peaks, and 1045 differentially bound Cp190 regions with lower occupancy in *CTCF^0* relative to WT were split into 1076 peaks. Accompanying the CTCF ChIP-seq, matches to the *Drosophila* CTCF motif MA0531.1 downloaded from the JASPAR website were indicated in all figures.

**Hi-C library preparation.** 60 third-instar larval CNSs (~600,000 cells) per biological replicate were dissected in ice-cold PBS. CNSs or a single whole-bodied female fly were crushed in RPMI supplemented with 10% fetal bovine serum using a micro-pestle. Cells were fixed in 1% (v/v) paraformaldehyde for 10 min at room temperature. The Hi-C libraries were prepared using MboI and MseI as restriction enzymes. Restricted ends were marked with biotin, then ligated. Fragmented DNA was enriched for pairwise DNA junctions by biotin pull-down using Dynabeads MyOne Streptavidin T1 beads following the manufacturer's instructions. Illumina sequencing libraries were prepared with standard protocols. 4 nM equimolar pools of multiplexed Hi-C libraries were subjected to paired-end sequencing on Illumina HiSeqX Ten and HiSeq4000 instruments.

**Hi-C data processing.** We pre-computed a table containing the positions of all restriction sites used for Hi-C present in the dm6 genome. The FASTQ read pairs were analyzed with a Perl script available for download in the Micmap[63] package (see Code Availability) to locate and separate fusion sites using the patterns /GATCGATC/, /TTATAA/, /GATCTAA/ and /TTAGATC/. The maximal length of each read was trimmed at 60 nucleotides, then reads were mapped to the dm6 genome using Micmap and matched to their closest pre-computed genomic restriction site. Read pairs were discarded if they (1) mapped to non-unique positions in the reference genome, (2) had indels or >2 mismatches per read, (3) represented fusion of 2 oppositely oriented reads within 2 kb of each other, which may have not resulted from ligation of 2 digested fragments (these fragments were used to estimate local copy number status of the underpinning genomic region), (4) were likely additional copies of a given read pair, i.e., likely PCR duplicates. Only chromosomes 2, 3, 4, and X were considered.

To assess the correlation of biological replicates, samples were downsampled to 45 million contacts per replicate. Raw Hi-C contact matrices were created by binning Hi-C pairs at 10 kb resolution. These matrices were then normalized with the ICE normalization implemented in iced v0.5.2[67]. Low coverage regions (bins with no contacts and those with the 2% smallest total number of contacts among bins) were filtered out. Pearson correlation coefficients were determined for every pair of normalized matrices by flattening each matrix and evaluating the Pearson correlation coefficient for the resulting vector, using only pairs of bins at a genomic distance below 1 Mb. The limitation on the distance was introduced to compare contacts at a scale relevant to the analyses performed in this manuscript which were at the level of CDs. Resulting Pearson correlation coefficients were ≥0.949 for all replicates, showing that they were well correlated and that WT and *CTCF^0* Hi-C matrices were globally similar. For the analyses presented in the main figures, pooled replicates of the same genotype were downsampled to 200 million contacts per genotype. Raw Hi-C contact matrices obtained by binning Hi-C pairs at 2 kb resolution were then normalized with the ICE normalization implemented in iced v0.5.2[67]. Low coverage regions (bins with no contacts and those with the 2% smallest total number of contacts among bins) were filtered out before normalization (these regions are marked by gray lines in Hi-C maps shown in the figures).

For each normalized Hi-C contact matrix, CD boundaries were called using TopDom[68]. Given a window size *w*, a physical insulation score was defined for each bin *i* as:

$$\log2 \frac{binSignal_i}{\sum_{i-w/2 < j < i+w/2} binSignal_j} \tag{1}$$

where $binSignal_i$ is the average normalized Hi-C contact frequency between *w* bins upstream of bin *i* and *w* bins downstream of bin *i* determined by TopDom. The strength of a boundary at bin *i* was thus estimated as the $\log_2$ of the $binSignal$ value at bin *i* normalized by its local average on a window of size *w*. With this definition, lower insulation scores indicate stronger boundaries. We extracted CD boundaries and physical insulation scores for Hi-C matrices at 2 kb resolution using window sizes 20, 40, 80, and 160 kb. CD boundaries found with all window sizes were merged, and the average insulation score obtained with all window sizes was retained. To facilitate comparisons of CD boundaries found in WT and *CTCF^0* genotypes and avoid mismatches due to small fluctuations of CD boundary positions obtained with different window sizes or genotypes, groups of consecutive boundaries (i.e., within 2 kb of each other) were merged. Groups of consecutive boundaries were replaced by the boundary with the lowest insulation score (average of both genotypes for boundaries common to WT and *CTCF^0*).

Hi-C maps were visualized in R and Juicebox[69] (see Supplementary Table 3 for links to interactive maps for browsing).

**A/B compartment calling**. A/B compartment calling was performed following the method proposed in Lieberman Aiden et al.[70]. Each individual chromosome arm (chr2L, chr2R, chr3L, chr3R, chr4, chrX) was analyzed separately. Normalized Hi-C contact matrices at 2 kb resolution were considered after discarding invalid bins (low coverage regions) and bins around centromeres (chosen for exclusion as dm6 coordinates >22,170,000 for chr2L, <5,650,000 for chr2R, >22,900,000 for chr3L, <4,200,000 for chr3R). Observed-over-expected matrices were generated by dividing the normalized Hi-C contact matrices by the average number of normalized Hi-C contacts at the corresponding genomic distance. For each chromosome arm, the first eigenvector of the correlation matrix was obtained by principal component analysis of the observed-over-expected matrix. Each eigenvector was then centered around zero by subtracting its mean value, then multiplied by the sign of the Pearson correlation between the eigenvector and the number of expressed gene TSSs per 2 kb bin. 2 kb bins with positive eigenvector values were assigned to compartment A, those with negative eigenvector values were assigned to compartment B. chr4 eigenvectors appeared to reflect a large-scale structure that separated the chromosome into two halves, and were thus excluded from Supplementary Fig. 3d.

**Comparison with CD boundaries from other Hi-C studies**. To assess whether CD boundaries called in our study could correspond to small CDs resolved in higher resolution Hi-C contact maps (analyzed at 500 bp resolution instead of 2 kb used here), we compared our CD boundary calls to CD coordinates published by Eagen et al[14]. and Ramírez et al.[17] (converted from dm3 to dm6 genome coordinates using the liftOver tool http://genome.ucsc.edu/cgi-bin/hgLiftOver) in Kc167 tissue culture cells. We counted how many small (≤4 kb) CDs identified in those published studies were close (within 2 kb) to one of our CD boundaries. We could have potentially mis-called such small domains as a domain boundary. The result is that Eagen et al. did not report CDs smaller than 6 kb. Only 31 of our domain boundaries were within 2 kb of a ≤ 4 kb CD identified by Ramírez et al. Thus, very few (31/3970, or <1%) of our domain boundaries may correspond to a small domain defined by Ramírez et al. We next asked: How many domain boundaries that disappear in $CTCF^0$ mutants could correspond to small domains? The result is that very few (4/567, or <1%) of our domain boundaries identified only in WT were within 2 kb of a ≤4 kb CD identified by Ramírez et al. Domain boundaries identified by Ramírez et al. are displayed together with domain boundaries identified in this study in all Hi-C screenshots throughout the manuscript for comparison.

**RNA-seq on larval CNSs**. WT, $CTCF^0$ and $Cp190^{KO}$ mutant third instar larval brains were dissected in ice-cold PBS. For RNA isolation, triplicates of 60 larval brains each were homogenized in TRIzol LS (ThermoFisher) with pestles (VWR) on ice. RNA was extracted following the manufacturer's instructions, remaining DNA digested with DNase I (Roche), and RNA was purified using RNAClean XP beads (Beckman Coulter). Strand-specific mRNA-seq libraries were prepared from 1 μg of total RNA after mRNA selection with NEBNext Oligo d(T)25 beads, using the NEBNext Ultra directional RNA library prep kit for Illumina following the manufacturer's instructions. Multiplexed libraries were sequenced on one lane of a HiSeq2500 (100 bp paired-end for $CTCF^0$ and WT control) or a Hiseq4000 (150 bp single-end for $Cp190^{KO}$ and WT control).

**Differential RNA-seq analysis**. RNA-seq reads were mapped both to the dm6 *Drosophila melanogaster* reference genome and to Flybase gene models and transcripts (dmel-all-r6.26.gtf.gz) using Micmap[63]. The results of both mappings were combined into spliced alignments in BAM file format. Then, htseq-count (v0.9.1) was used to produce read counts per gene[71]. Statistical analysis was performed in R (v3.5.1). Genes with <1 count per million in at least three replicate samples were filtered out using EdgeR (v3.22.5)[66]. Normalization and differential expression analysis were performed in DEseq2 (v1.22.1)[72] individually for both WT versus $CTCF^0$ and WT versus $Cp190^{KO}$ samples. Statistical significance was tested by Wald test and the Benjamini-Hochberg method was used for multiple testing adjustment. A significance threshold of |fold change| > 1.5 and p-adjusted < 0.05 was used to identify DE genes. The R package ggplot2 (v3.2.1) was used for data visualization.

**RNA-FISH**. Labeled RNA probes were generated by in vitro transcription with Dig-UTP labeling mix (Roche 11277073910) and T7 RNA polymerase (Roche 10881767001) antisense to full-length complementary DNA clones of *SP1029* (FI20034) and *IFT52* (MIP14443), genomic DNA amplified from dm6 coordinates chr3L: 10263888-10266244, or cDNAs amplified using gene-specific primers from a cDNA library prepared from *Drosophila* embryos (see Supplementary Data 10 for primer sequences). After DNase I digestion for 20 min at 37 °C, probes were fragmented by incubating 20 min at 65 °C in 60 mM $Na_2CO_3$, 40 mM NaHCO₃ pH 10.2, precipitated in 300 mM sodium acetate pH 5.2, 1.25 M LiCl, 50 mg/ml tRNA and 80% EtOH, resuspended in 50% formamide, 75 mM sodium citrate pH 5, 750 mM NaCl, 100 μg/ml salmon sperm DNA, 50 μg/ml heparin and 0.1% Tween20, and stored at −20 °C. Embryos or third instar larval cuticles were fixed in

4% paraformaldehyde for 30 min at room temperature, washed, and then stored in 100% MeOH at −20 °C for at least overnight. Samples were rehydrated in PBS with 0.1% Tween20, post-fixed in 4% paraformaldehyde for 20 min at room temperature, progressively equilibrated to hybridization buffer (50% formamide, 75 mM sodium citrate pH 5, 750 mM NaCl) and heated to 65 °C. RNA probes were diluted 1:50 in hybridization buffer, denatured at 80 °C for 10 min then placed on ice, and added to the samples overnight shaking at 65 °C. Samples were washed 6 times 10 min in hybridization buffer at 65 °C, then progressively equilibrated to PBS with 0.1% Triton X-100. Samples were incubated overnight at 4 °C in anti-dig peroxidase (Roche 11207733910) diluted 1:2000 in PBS, 0.1% Triton X-100, 1× Western blocking reagent (Sigma 1921673). Samples were washed six times 10 min in PBS with 0.1% Tween20, labeled with Cyanine 3 tyramide in the TSA Plus kit (Perkin Elmer NEL753001KT) for 3 min at room temperature, washed 6 times 10 min in PBS with 0.1% Tween20, and finally mounted with DAPI to stain DNA. Images were acquired on a Zeiss LSM 880 microscope with a ×20 objective and visualized with Fiji software v2.1.0/1.53c.

**Insulator reporter**. An insulator reporter (Fig. 4a) was designed with an enhancer (*OpIE2*) equidistant from EGFP and mCherry fluorescent reporters with basal *Hsp70* promoters. A *gypsy* insulator is present in the reporter plasmid, downstream of the EGFP transcription unit. Selected CTCF binding sites (Supplementary Fig. 4c) were PCR-amplified from genomic DNA and cloned in between the enhancer and EGFP. Control reporters had a neutral spacer (a fragment of the bacterial *Kanamycin* resistance gene) or the *gypsy* insulator in between the enhancer and EGFP. In addition, one CTCF binding site (fragment N) was mutagenized by PCR to mutate 2 bp in a CTCF motif (ATGTCAGAGGGCGCT converted to ATGTCAGACAGCGCT). All plasmids were transfected in parallel into S2 cells (originally purchased from ATCC, reference number CRL-1963) in triplicates in a 96-well plate using 100 ng of reporter plasmid and Effectene (QIAGEN) following the manufacturer's instructions. After 48 h, fluorescence was measured on a NovoCyte Flow Cytometer (ACEA) using FITC and PE-TexasRed detection settings. Recordings were gated to discard measurements of untransfected cells (Supplementary Fig. 4a). Distributions of mCherry/EGFP fluorescence ratios in thousands of single transfected cells were plotted and the median mCherry/EGFP ratio was extracted for each experiment. The average of these median values obtained for each replicate is plotted in Fig. 4c as a function of the total CTCF ChIP-seq read counts in S2 cells on the cloned fragment tested in the insulator reporter—extracted using bedtools multicov[73] applied to CTCF ChIP-seq data in S2 cells[42] (GEO accession GSM1015410).

**Recombinant protein pull-downs**

*Purification of N-terminal CTCF constructs.* The sequence encoding WT or Y248A F250A mutant versions of the dmCTCF N-terminus (residues 1-293) were cloned into a pET-based vector with an N-terminal GFP-tag and a C-terminal His₆ tag. The constructs were transformed into an *E.coli* expression strain (Rosetta), and 1 liter cultures were grown in TB-medium to an OD(600) of 1.0 at 37 °C. The culture temperature was then reduced to 18 °C and IPTG was added to a final concentration of 0.5 mM. Cells were harvested after overnight incubation at 18 °C by centrifugation, and the cell pellet was resuspended in 2 volumes of Lysis Buffer (50 mM Tris pH 7.5, 300 mM NaCl, 5 % glycerol, 25 mM Imidazole). Cells were opened by sonication, and the lysate was clarified by centrifugation at 50,000 × g at 4 °C. The supernatant was loaded onto a 5 ml HisTrap column (GE Healthcare), washed extensively with Lysis Buffer, and the bound material was eluted with Lysis Buffer supplemented with 400 mM Imidazole. The eluate was then diluted 10-fold with buffer QA (20 mM Tris pH 7.5, 100 mM NaCl, 5% glycerol), and the resulting solution was loaded onto a 5 ml HiTrap-Q column (GE Healthcare). After washing the column with 5 column volumes (cV) of QA buffer, the bound material was eluted with a 5 cV gradient from QA to QB (20 mM Tris pH 7.5, 1000 mM NaCl, 5% glycerol). Fractions containing the CTCF protein at sufficient purity were identified by SDS-PAGE followed by Coomassie staining. Proteins aliquots were snap-frozen in liquid nitrogen and stored at −80 °C.

*Purification of SA-Vtd complex.* The sequences encoding dmSA (residues 102–1085) and Vtd (Rad21) (residues 273-458) were cloned into a pET-based vector with an N-terminal His₁₀-TwinStrep-3C tag on SA. The complex was expressed in 1 liter of *E.coli* (Rosetta) grown in TB. Growth, induction of expression, and cell harvesting and lysis were carried out as described for CTCF constructs. Clarified lysates were loaded onto a 5 ml StrepTrap column (GE Healthcare), washed with 5 cV of Lysis buffer, and bound material was eluted with 8 cV of elution buffer (20 mM Tris pH 7.5, 100 mM NaCl, 5 % glycerol, 2.5 mM des-thiobiotin). The eluate was loaded on a 5 ml HiTrap-Q column (GE Healthcare), and after washing the column with 5 column volumes (cV) of QA buffer, the bound material was eluted with a 5 cV gradient from QA to QB (20 mM Tris pH 7.5, 1000 mM NaCl, 5% glycerol). Fractions containing the purified SA-Vtd complex were identified by SDS-PAGE and Coomassie staining, pooled, aliquoted, snap-frozen in liquid nitrogen, and stored at −80 °C.

*Pulldowns between CTCF and SA-Vtd.* Proteins were diluted to a final concentration of 2.5 μM in 500 μl of binding buffer (20 mM Tris pH 7.5, 150 mM

potassium acetate, 10 % glycerol) and allowed to bind to each other at 4 °C for 2 h. Twenty microliters of this solution was removed as 'input' sample and boiled in SDS-PAGE loading buffer. GFP-binder beads (Agarose beads covalently bound to GFP-nanobody; 20 μl per reaction) were washed in binding buffer and added to the binding reactions for 30 min at 4 °C on a rotating wheel to bind to the GFP-tagged CTCF construct. Beads were harvested by centrifugation (1 min, 700 × g) and washed twice with 1 ml of binding buffer. The final immobilized material was eluted by boiling in 50 μl of SDS-PAGE loading buffer. Inputs and pulldowns were loaded onto a 12% SDS-PAGE gel, and the proteins were visualized by staining with Coomassie.

*Pull-downs between C-terminal CTCF constructs and Cp190 BTB domain.* Expression plasmids encoding GFP-His-tagged constructs of the C-terminal domain of CTCF (all with Ampicillin resistance) were co-transformed with an expression plasmid carrying a His-tagged Cp190 BTB-domain (with Kanamycin resistance) into the *E.coli* Rosetta strain. Colonies were inoculated in 10 ml TB cultures and grown at 37 °C to an OD(600) of 1. The culture temperature was then reduced to 18 °C, and 0.5 mM IPTG was added to induce protein expression. Cells were harvested after overnight incubation at 18 °C, and the pellets were resuspended in 2 volumes of lysis buffer (50 mM Tris pH 7.5, 200 mM NaCl, 5% glycerol, 25 mM Imidazole). Cells were lysed by sonication and the lysate was clarified by centrifugation at 16000 g for 10 min at 4 °C. The lysates were split into two halves, which were incubated for 1 h at 4 °C with either 20 μl of GFP-binder resin or 20 μl of Ni(2+)-NTA resin, to pull down only CTCF-constructs or both CTCF and CP190-BTB, respectively. The beads were then washed three times with 1 ml of Lysis buffer to remove non-specifically bound proteins. The bound material was eluted either by boiling in SDS-loading buffer (for GFP pulldowns) or by incubation with Lysis buffer supplemented with 500 mM Imidazole (for Ni (2+)-NTA pulldowns), and analysed by SDS-PAGE followed by Coomassie staining.

**Co-purification of CTCF interactors from embryo nuclear extracts.** Soluble nuclear protein extracts were prepared from WT (OregonR) 0–14 h embryos. Thirty grams of embryos were dechorionated, taken up in 30 ml of NU1 buffer (15 mM HEPES pH 7.6, 10 mM KCl, 5 mM $MgCl_2$, 0.1 mM EDTA pH 8, 0.5 mM EGTA pH 8, 350 mM sucrose, 2 mM DTT, 0.2 mM PMSF), and dounce-homogenized. The lysate was filtered through a double layer of miracloth, then centrifuged 15 min at 9000 rpm at 4 °C. The nuclei pellet was resuspended and lysed in 30 ml of high-salt buffer (15 mM HEPES pH 7.9, 400 mM KCl, 1.5 mM $MgCl_2$, 0.2 mM EDTA, 20% glycerol, 1 mM DTT, protease inhibitor cocktail) rotating for 20 min at 4 °C, and ultracentrifuged 1 h with a SW40 rotor at 38000 rpm at 4 °C. The lipid layer was removed by suction and the soluble nuclear extract was dialyzed into 15 mM HEPES pH 7.9, 200 mM KCl, 1.5 mM $MgCl_2$, 0.2 mM EDTA pH 7.9, 20% glycerol, 1 mM DTT with a 6-8 kDa molecular weight cut-off membrane. Soluble nuclear extract was snap-frozen in liquid nitrogen, and stored at −80 °C. *Drosophila* CTCF[1–293] fused to an N-terminal GFP-3C tag and a 3C-$His_6$ C-terminal tag was purified from bacterial lysates by Ni-NTA affinity then ion-exchange chromatography as described above. Purified GFP-3C-CTCF[1–293]-3C-$His_6$ was immobilized on GFP binder beads, of which 30 μl bead volume were then incubated with 6 mg of *Drosophila* embryo nuclear extract in a total volume of 10 ml of IP buffer (50 mM Tris-Cl pH 7.5, 150 mM potassium acetate, 2 mM $MgCl_2$, 10% glycerol, 0.1 mM DTT, 0.2% Igepal, 1× complete protease inhibitor cocktail) rotating for 3 h at 4 °C. Beads were washed three times with IP buffer, rotating for 10 min at 4 °C for each wash. Proteins were eluted with 3 C protease, adjusted to 1× SDS-loading buffer and loaded on an SDS-PAGE gel. A duplicate experiment was similarly performed with nuclear protein extracts prepared from another biological replicate embryo sample. Peptides covering the entire CTCF full-length protein were recovered, indicating that pull-downs with CTCF N-terminus recovered interactors of full-length CTCF.

**Mass spectrometry analysis.** Protein samples were separated by SDS-PAGE and stained by Coomassie. Gel lanes between 15–300 kDa were excised into five pieces and digested with sequencing-grade trypsin. Extracted tryptic peptides were dried and resuspended in 0.05% trifluoroacetic acid, 2% (v/v) acetonitrile. Tryptic peptide mixtures were injected on a Dionex RSLC 3000 nanoHPLC system (Dionex, Sunnyvale, CA, USA) interfaced via a nanospray source to a high-resolution mass spectrometer LTQ-Orbitrap Velos Pro. Peptides were loaded onto a trapping microcolumn Acclaim PepMap100 C18 (20 mm × 100 μm ID, 5 μm, Dionex) before separation on a C18 reversed-phase custom-packed column using a gradient from 4 to 76% acetonitrile in 0.1 % formic acid. In data-dependent acquisition controlled by Xcalibur software (Thermo Fisher), the 10 most intense multiply charged precursor ions detected with a full MS survey scan in the Orbitrap were selected for collision-induced dissociation (CID, normalized collision energy NCE = 35%) and analysis in the ion trap. The window for precursor isolation was of 4.0 *m/z* units around the precursor and selected fragments were excluded for 60 s from further analysis. Data files were analyzed with MaxQuant 1.6.3.4 incorporating the Andromeda search engine[74,75] for protein identification and quantification based on IBAQ intensities[76]. The following variable modifications were specified: cysteine carbamidomethylation (fixed) and methionine oxidation and protein N-terminal acetylation (variable). The sequence databases used for searching were *Drosophila*

*melanogaster* and *Escherichia coli* reference proteomes based on the UniProt database (www.uniprot.org, versions of 31 January 2019, containing 21,939 and 4915 sequences respectively), and a contaminant database containing the most usual environmental contaminants and the enzymes used for digestion (keratins, trypsin, etc). Mass tolerance was 4.5 ppm on precursors (after recalibration) and 0.5 Da on CID fragments. Both peptide and protein identifications were filtered at 1% FDR relative to hits against a decoy database built by reversing protein sequences. The MaxQuant output table proteinGroups.txt was processed with Perseus software[77] to remove proteins matched to the contaminants database as well as proteins identified only by modified peptides or reverse database hits. Next, the table was filtered to retain only proteins identified by a minimum of two peptides, the IBAQ quantitative values were log-2 transformed and missing values imputed with a constant value of 9.

**Generation of $Cp190^{KO}$ animals.** We cloned ~1.5 kb homology arms (dm6 coordinates chr3R:15276111-15274519 and chr3R:15271056-15269404) into the pHD-DsRed-attP vector[78]. Guide RNAs close to the start and stop codons of the *Cp190* open reading frame were cloned into pCFD3 vector[79]. Plasmids were co-injected into *nanos-Cas9* embryos[79]. Experiments were performed in animals transheterozygous for two independent knockout alleles.

**Generation of $Cp190^0$ animals.** $Cp190^{KO}$ mutants were rescued into viable and fertile adults with an FRT-flanked 7 kb *Cp190* genomic rescue transgene (dm6 coordinates chr3R:15269425-15276409) amplified by PCR. The *Cp190* rescue cassette was excised from male and female germlines through *nanos-Gal4:VP16* (NGVP16)-driven expression of *UAS-FLP*. $Cp190^0$ animals were collected from crosses between such males and females.

**Statistics and reproducibility.** All described replicate experiments are biological (not technical) replicates. For all box plots: center line, median; box limits, upper and lower quartiles; upper whisker extends to the largest value no further than 1.5× interquartile range from the upper hinge; lower whisker extends to the smallest value no further than 1.5× interquartile range from the lower hinge; points, outliers. Figure 2g: This experiment was repeated twice from independently grown bacterial cultures, with similar results. Figure 3e and Supplementary Fig. 1a–b: *n* = 10 independent third instar larvae per genotype were examined over two independent experiments each. All animals showed similar expression patterns for a given gene, that was characteristic of each genotype. RNA-FISH probes for additional genes were tested on larval nervous systems but discarded because they showed an inconsistent pattern (variable, asymmetric signal in the optic lobes in all genotypes) that we concluded was non-specific background. Figure 6c, e: *n* = 50 independent embryos per genotype were examined over two independent RNA-FISH experiments each. All animals showed similar expression patterns for a given gene, that was characteristic of each genotype. Supplementary Fig. 2a: The experiment was repeated twice with independently prepared extracts, with similar results. Supplementary Fig. 5b: The pull-down experiments were repeated twice from independently grown bacterial cultures, with similar results.

**Reporting summary.** Further information on experimental design is available in the Nature Research Reporting Summary linked to this paper.

## Data availability
All sequencing data (Hi-C, ChIP-seq, RNA-seq) that support the findings of this study were deposited in Gene Expression Omnibus with accession code GSE146752. Hi-C maps are browsable on Juicebox (links in Supplementary Table 3). Mass spectrometry proteomics data were deposited to the ProteomeXchange Consortium via the PRIDE partner repository with the dataset identifier PXD019487. All other relevant data supporting the key findings of this study are available within the article and its Supplementary Information files or from the corresponding author upon reasonable request. Additional information is provided in Supplementary Data files 1–10 and a reporting summary for this Article is available as a Supplementary Information file. Source data are provided with this paper.

## Code availability
All software used as described in the Methods to map, visualize and analyze data is published open source and freely available for download in the following links: "Micmap v2.20200223 [https://github.com/sib-swiss/micmap]"; "DESeq2 v1.22.2 [https://bioconductor.org/packages/release/bioc/html/DESeq2.html]"; "HTSeq v0.9.1 [https://github.com/simon-anders/htseq]"; "iced v0.5.2 [https://github.com/hiclib/iced]"; "TopDom v0.0.2 [https://github.com/jasminezhoulab/TopDom]"; "R v3.5.1 [https://www.R-project.org/]" with packages "csaw v1.16.1 [https://bioconductor.org/packages/release/bioc/html/csaw.html]", "edgeR v3.22.5 [https://bioconductor.org/packages/release/bioc/html/edgeR.html]", "Eulerr v6.0.0 [https://cran.r-project.org/package=eulerr]" and "ggplot2 v3.1.0 [https://ggplot2.tidyverse.org/]"; "bedtools multicov v2.29.2 [https://bedtools.readthedocs.io/en/latest/]"; "Juicebox v1.5.1 [aidenlab.org/juicebox]". Custom scripts are provided in "link [https://github.com/gambettalab/kaushal2020/]".

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

## Acknowledgements

We thank Winship Herr, Richard Benton, Jean-Yves Roignant and Naoko Mizuno for critical comments on the manuscript. We thank Patrice Waridel and Manfredo Quadroni for mass spectrometry analyses. We thank René Dreos for advice on statistical analyses. MCG thanks Eileen Furlong for support during early phases preceding this work. Deep sequencing was performed at the Genomic Technologies Facility (GTF), mass spectrometry was performed at the Protein Analysis Facility (PAF) and imaging was performed at the Cellular Imaging Facility (CIF) at the Center for Integrative Genomics, Faculty of Biology and Medicine, University of Lausanne, Switzerland. This work was supported by the Swiss National Science Foundation (SNSF #184715 to M.C.G.) and the University of Lausanne.

## Author contributions

M.C.G. conceived the project and designed experiments. E.L.A. and A.O. conceived and designed Hi-C experiments. A.K., G.M., I.O., A.O., M.T., P.C., A.S., and M.C.G. performed the experiments. J.D., P.C., A.S., O.D., F.M., C.I., Y.E., D.W., M.S.S., and N.G. analyzed data. Y.E. created links interactive browsing of Hi-C and ChIP-seq data in Juicebox. M.C.G. prepared the manuscript with input from all authors.

## Competing interests

The authors declare no competing interests.
