## [Peer Review File · Nature Communications]

REVIEWER COMMENTS

Reviewer #1 (Remarks to the Author):

The manuscript by Gambetta MC, Lieberman-Aiden E and collaborators with the title: "CTCF loss has limited effect on genome architecture in Drosophila despite critical regulatory functions" report the effects of CTCF elimination on genome architecture in Drosophila. The results illustrate that a large portion of the fly domain boundaries are not sensitive to Ctcf mutants, in contrast to what is known in vertebrates where CTCF is a key component of the topological domains in association with cohesins. In addition this investigation supports a regulatory participation of CTCF at the gene expression level, in particular, in the central nervous system.

This is a relevant piece of work, technically sounding that address an unsolved aspect that has to do with the role of CTCF in vivo, and in particular in the CNS. The results confirm a series of previous observations and with different strategies the authors ask diverse functions of CTCF. The manuscript lack a flow of justifications and interpretations that may link each of the aspects addressed in the present work. For example, it would be interesting to further characterize the phenotype observed in the absence of CTCF in the CNS. The interaction with CP190 was already predicted and deeper discussion is needed. Another aspect is the effect or not has CTCF over the nearby genes. Together, the manuscript has a real potential but some aspects need to be reanalyzed and complemented.

Specific comments

- In: "Recent studies have for example perturbed specific contact domains in flies to address their biological roles without knowing whether they are CTCF-mediated or compartmental, which is a key point as different types of contact domains may have different regulatory properties" the following references should be included: <https://www.nature.com/articles/s41586-019-1035-4?draft=marketing> and <https://www.nature.com/articles/s41467-020-14651-z>.

- It would be convenient to complement the knock-out phenotype study and rescue experiments by examining the levels of CTCF in the cells tested, at least by immunofluorescence.

- There is a discrepancy that needs to be clarified in Fig. 1a, 1b and Suppl. Movie 1. Some CTCFKO mutants (60%) hatch into adults with spasmodic movements suggesting a neurological phenotype that might be the cause of their short lifespan (Figs. 1a, 1b, Supplementary Movie 1) in contrast to "CTCF0 mutants never hatch from the pupal case (Fig. 1c)".

- On Fig. 2C it could be useful to visualize together the insulator scores of wt and KOs and to incorporate a heatmap outlining the differences between them.

- Another inconsistency is found between a conclusion stated in the text and the title of the manuscript: "These results strongly suggest that CTCF mediates formation of physical boundaries" versus the title: "CTCF loss has limited effect on genome architecture in Drosophila despite critical regulatory functions".

- It is necessary to analyze the CTCF distribution and contribution to CTCF-dependent and independent boundaries in the co-existence of the so-called "architectural proteins" like BEAF-32, CP190, Su(Hw), among others. This is relevant since it may exist additional sub-classes of boundaries associates to CTCF and other TFs and/or even co-factors.

- In line with the previous point, it is important to try to determine why some CTCF sites seem to be more prominent in comparison to others, including the author's proposition that CTCF can reinforce boundaries redundantly. Is it relevant to discuss here which are the alternative CTCF partners and mechanisms of action ?

- CTCF distribution is explained taking advantage of a "forced" interaction with cohesion obtaining the same results as the one already published, in which cohesin interaction is not critical in fly. How this results contribute to set of results presented in the manuscript ?

- To some extent it is not surprising that CTCF affects nearby genes. This is because the location genome-wide of CTCF in flies has shown significant proximity to TSSs. Author's address this point but it would be relevant to analyze this aspect in more detail. Mutations can be introduced in CTCF sites located distal or proximal to a selected CNS genes and determine their differential effects, if there is any ?

- Among the 400 genes with expression changes, a compartment analysis may help to better correlate the gene expression changes and modifications in the spatial chromatin configuration.

- In the enhancer-blocking series of experiments it would be relevant to define or select some of the CTCF sites based on their differential locations in the genomes (in proximity or distal to TSSs) and test them using the same assay. In addition and using an average sequence of more or less 360 bp it could be interesting to see if there are other adjacent binding sites for additional transcription factors, in particular, architectural proteins or even novel factors or co-factors.

- It is not surprising to see that CP190 and CTCF are partners since it has already been demonstrated their co-localization. Therefore, authors should exploit deeply the genes that are commonly altered when those two factors are mutated and if they generate a particular phenotype in the fly.

Reviewer #2 (Remarks to the Author):

In this MS the authors present data demonstrating that a limited number of TAD borders in *Drosophila* cells depend on the binding of CTCF. Of note, most of TAD borders in *Drosophila* cells do not depend on CTCF, and this is the first study directly demonstrating this via Hi-C analysis of CTCF0 cells. The authors also show that CTCF controls expression patterns of specific genes, but weakening of TAD boundaries at former CTCF peaks in CTCF0 mutants is not a mere consequence of altered transcription. Finally, it is shown that in some locations CTCF cooperates with Cp190. This is a solid experimental study that merits criteria of originality and general importance. I can recommend publication in *Nature Communications* after a revision. The following issues are to be addressed:

1. The results obtained suggest that several architectural proteins may simultaneously contribute to the formation of a TAD boundary. In CTCF0 cells some of the boundaries at former CTCF peaks were partially retained (Supplementary Fig 1a). When insulator score was analysed in transient transfection experiments (scatter plot in Fig. 4a) two groups of CTCF test fragments can be easily observed (above the spacer level): C-D-G-F, and A-M-B-H-N-E. It would appear that the dependency between the insulator strength and ChIP-seq counts is the same for both groups, but fragments from CDGF are much more strong insulators than the AMBHNE albeit demonstrate the same CTCF occupancy according the ChIP-seq. Taking all this into consideration it would be important to perform additional ChIP-seq experiments with antibodies against known *Drosophila* architectural proteins.

2. To clarify the cooperation between CTCF and Cp190 it will be important to perform Hi-C analysis on cells from Cp190-KO animals

3. Related to the ability of dCTCF to insulate neighbouring loci: it would be insightful to compare the physical insulation score profile around CTCF peaks and sites of other "architectural" proteins such as BEAF-32, Su(Hw) etc. in WT flies.

4. I suppose the authors should describe their findings on CD borders in more details in the main text: how many borders were identified in WT and in the CTCF-0, and how many borders lost in the CTCF-0 were bound with CTCF in WT.

5. The authors state:

"The vast majority of strongly affected domain boundaries in CTCF0 mutants occurred at former

CTCF peaks (Fig. 2d)".

The authors should show a completely marked vertical axis in Fig. 2d, because in the current version it is impossible to evaluate the actual range of insulation defects at CTCF-bound CD borders "strongly affected" in CTCF-deficient flies.

6. The authors propose that "... CTCF is required to form physical boundaries with strengths generally proportional to its occupancy on DNA".

However, in my opinion, this is a broad statement. What do we actually see in the results?

(i) not all but only a minor fraction (what proportion exactly?) of CTCF-bound CD borders are lost in the absence of CTCF;

(ii) Figure 2d shows that the mostly affected CD borders are bound with CTCF, but does not allow one to accurately estimate the dependence between the IS defect value and CTCF occupancy at the border.

Thus, the authors should indicate the number of lost (or significantly weakened) CD borders in the CTCF-0, and provide a quantitative analysis of the relationships between CTCF occupancy and the degree of the IS defects at the CD borders.

5. The authors conclude that "...the pervasive weakening of physical boundaries observed at former CTCF peaks in CTCF0 mutants (Fig. 2f) is not a mere consequence of altered transcription". However, this statement is based on visual inspection of limited number of gene loci. Meanwhile, the interplay between transcription and CTCF-driven chromatin 3D organization is far from the complete description, and the unique model system presented in this work provides a good opportunity to get some novel insights. I suggest the authors could annotate spatial contacts between promoters of DE genes and nearby enhancers, and to investigate whether expression deregulation is caused by formation or disruption of loops with CTCF-bound regulatory elements.

Reviewer #3 (Remarks to the Author):

The manuscript describes the role of CTCF in 3D genome architecture in *Drosophila*. Authors make use of CTCF mutants and Hi-C to explore the role of this protein in the establishment of contact domains and enhancer-promoter interactions. Authors also find that a second protein, CP190, interacts with CTCF and is involved in the establishment of 3D architecture.

The manuscript addresses an important issue in the field of nuclear architecture and its relationship to gene expression. In spite of its evolutionary conservation, CTCF appears to function in a different manner in *Drosophila* and vertebrates. Although the authors do not completely answer the question, the observations go a long way in offering insights into how *Drosophila* and vertebrates appear to use slightly different mechanisms to organize their genomes in the 3D nuclear space. The following are a few comments that may help the authors prepare a revised version of the manuscript.

1. Authors should include a Supplementary Table with QC results of Hi-C libraries, such as those obtained using Juicer to process the data. In particular, information on intra- versus inter-chromosome interactions and contacts <20 kb and >20 kb would help the reader judge the quality of the Hi-C data.

2. 825 CTCF peaks seems like a low number. Was the CTCF antibody described previously and its specificity tested? Please include either a description of previously published information or details on the characterization of the antibody

3. "These results strongly suggest that CTCF mediates formation of physical boundaries". Previous work has shown that, what appear to be boundaries between domains observed using low resolution Hi-C, of the same order of contacts employed here, are in fact small domains. It is possible that what appears to be physical insulation is caused by compartmental interactions between small active compartments flanking large compartments containing no genes or silenced genes. One possibility to distinguish between the two possibilities would be to compare the Hi-C

data described here with that obtained in cultured cells by, for example, Eagen et al 2017 or Rowley et al 2018, which contain around 1 billion valid contacts. This comparison could be performed in regions of the genome containing the same transcribed genes and CTCF peaks. This issue is conceptually important i.e. does *Drosophila* CTCF mediate compartmental interactions between active domains or does it create boundaries by some unknown mechanism that does not involve the formation of loops like in mammals?

4. Figure 2g. Although the results in this figure look very clean and convincing, they are difficult to rationalize. The YDF residues of human CTCF interact with a pocket formed by SA2 and SSC1, and several residues of SSC1, which is not present in the experiments described in Figure 4g, are necessary for the interaction. Authors should comment on this.

5. Figure 3. It has been shown that some CTCF sites in *Drosophila* co-localize with SuHw whereas others do not. Since SuHw has been shown to repress transcription, at least under some circumstances, it would be interesting to examine whether there is a correlation between up or down-regulation of transcription in CTCF mutants and the presence or absence of SuHw at a subset of sites. Since SuHw has a well-defined binding motif, it may be sufficient to look for the motif adjacent to CTCF sites without having to perform ChIP-seq.

6. Figure 4. Authors should explain the hypothesis for how the test fragments containing CTCF sites would affect enhancer-promoter interactions in the experiments described in Figure 4. The plasmids do not integrate in the chromosomes and are presumed to be circular. Therefore, in principle, the enhancer could contact the promoter in either direction around the plasmid. Perhaps having a second site to insert the test fragments would have made the results easier to interpret. Authors should discuss these possible shortcomings in the text. It is probably not appropriate to use results from these experiments to speculate what CTCF does in the normal situation. The conclusion "(4) Sites bound by CTCF do not directly repress or activate transcription, but rather functionally insulate promoters and enhancers in a reporter assay in S2 cells" in the Discussion is too strong.

7. "Cp190 colocalized with CTCF at nearly all (77%) CTCF peaks". At least for me, "nearly all" would be 99%. Please let the reader decide and delete "nearly all".

8. Figure 5f. It would be interesting if the authors came up with a more sophisticated model than that shown in Figure 5f. In mammals, the "boundary" effect of CTCF on E-P interactions can be explained by cohesin extrusion, which may increase the frequency of interactions within the loop but, by stopping at CTCF convergent sites, decreases interactions between regulatory sequences inside and outside of the loop. As a consequence, the CTCF site appears to form a boundary that interferes with E-P interactions. It would be surprising if cohesin is unable to extrude in *Drosophila*. However, the absence of puncta at *Drosophila* domains "flanked" by CTCF suggest that, even if there is extrusion by cohesin, this complex does not appear to stop at CTCF sites. Results from high resolution Hi-C in *Drosophila* cells suggest that the "boundaries" are in fact small active domains that interact with other active domains in the A compartment. It's interesting that CTCF and other *Drosophila* architectural proteins form large puncta in IF experiments (see for example Gerasimova et al 2008, where CTCF and CP190 are shown to co-localize at large dots in the nucleus). These puncta are similar to what would be considered membraneless organelles formed by LLPS. Proteins like CTCF and CP190 may work by increasing the frequency or stability among A compartmental domains. It would be interesting to put all these facts together into a congruent model.

Reviewer #4 (Remarks to the Author):

In this study authors report novel findings on the role of CTCF in chromosome folding and transcriptional regulation in *Drosophila melanogaster*. The manuscript is clearly written, experiments appear well conducted, presented and interpreted. I only have minor suggestions and support timely publication without the need for another round of revisions.

1. 10% of DE genes had a CTCF peak within ± 1 kb of their transcriptional start site (TSS). Is that equally distributed amongst up- and down- regulated genes?

2. When presenting the changes in CP190 occupancy in the CTCF0 mutant it would be helpful to show density heatmaps of CP190 ChIP-seq signal centered at CTCF peaks, in the WT and CTCF0 mutant side by side. Ideally in the main figures, as opposed (or in addition to) the Venn diagram of figure 5b.

3. It would be interesting if authors could speculate further about why corner peaks are not detected at CTCF-dependent contact domain boundaries, in light of the CTCF-SA-Vtd/Rad21 interaction being conserved in *Drosophila*. Do authors think it is a detection issue of the Hi-C, or something inherently different in how CTCF and cohesin cooperate in *Drosophila*? Do these findings imply that the CTCF-SA-Rad21 interaction in mammals is unlikely to be sufficient to create Hi-C peaks, and that additional (potentially mammalian-specific) actors must be involved downstream?

Elphege Nora

Point by point response to reviewer comments

REVIEWER 1

General comment 1

The manuscript lack a flow of justifications and interpretations that may link each of the aspects addressed in the present work. For example, it would be interesting to further characterize the phenotype observed in the absence of CTCF in the CNS.

Response to general comment 1

To address the reviewer's comment, **we summarize all datasets generated in WT and *CTCF*⁰ mutants (contact domain boundary defects, CTCF and Cp190 ChIP occupancy, mRNA-seq expression) in new Figure 5. This allows us to link physical insulation defects observed in *CTCF*⁰ mutants to differential Cp190 binding in *CTCF*⁰ mutants relative to WT.** These new findings are described in our response to point 7 further below.

We also provide a unified summary of our findings in a model (Fig. 6f). 3 types of CTCF-dependent boundaries (strictly CTCF-dependent, partially CTCF-dependent and CTCF-independent) are depicted with different structural (Fig. 2) and functional (Figs. 3 and 4) dependencies on CTCF. These boundaries also differ in how they are occupied by Cp190 occupancy in *CTCF*⁰ mutants (Fig. 5). This also depicts that CTCF and Cp190 co-bound at CTCF-dependent boundaries co-regulate a subset of genes (Fig. 6).

Our finding that CTCF expression in neural stem cells or neurons is essential for fly viability identified the nervous system as the most biologically relevant tissue in which to study CTCF function and explained why molecular phenotypes were subsequently analysed in dissected nervous systems. CTCF does not appear to preferentially regulate functionally related genes (Fig. 2) or affect specific cell lineages (Fig. 3e). Detailed anatomical and functional analyses of *CTCF*⁰ nervous systems were therefore unlikely to advance our understanding of CTCF function, and we instead focused on molecular phenotypes which could be compared to those reported in mammalian cells.

General comment 2

The interaction with CP190 was already predicted and deeper discussion is needed.

Response to general comment 2

As requested by the reviewer, we now discuss the original reports that led to current speculation about the relevance of the CTCF-Cp190 interaction. This highlights that prior assumptions were based on partial and in some cases inaccurate findings, and a comparison of *CTCF* and *Cp190* mutant phenotypes was overdue. The following text was added to the Discussion: **"The relevance of the CTCF-Cp190 interaction has been debated. On the one hand, Cp190 was assumed to be required for CTCF's insulator function based on the observations (1) that the enhancer-blocking activity of a *Hox* gene insulator in transgenic reporter assays depended on both CTCF and Cp190, and (2) that CTCF failed to be recruited to many sites on polytene chromosomes in *Cp190* mutants (Gerasimova et al., 2007; Mohan et al., 2007). The latter observation was, however, not reproduced in genome-wide ChIP experiments in Cp190 knock-down cells (Schwartz et al., 2012). On the other hand, no common CTCF and Cp190 target genes were known (Savitsky et al., 2016), and the interaction between CTCF and Cp190 was recently concluded to be dispensable *in vivo* (Bonchuk et al., 2015). The latter conclusion was based on deleting residues in CTCF that did not interact with Cp190 in our pull-down experiments (Supplementary Figure 5b). We identified genes with concordant transcriptional changes upon loss of either CTCF and Cp190 that are potentially directly regulated by both proteins."**

General comment 3

Another aspect is the effect or not has CTCF over the nearby genes.

Response to general comment 3

This point was addressed in the response to point 9.

Point 1

In: “Recent studies have for example perturbed specific contact domains in flies to address their biological roles without knowing whether they are CTCF-mediated or compartmental, which is a key point as different types of contact domains may have different regulatory properties” the following references should be included: <https://www.nature.com/articles/s41586-019-1035-4?draft=marketing> and <https://www.nature.com/articles/s41467-020-14651-z>.

Response to point 1

Following the reviewer’s helpful suggestion, the reference was added.

Point 2

It would be convenient to complement the knock-out phenotype study and rescue experiments by examining the levels of CTCF in the cells tested, at least by immunofluorescence.

Response to point 2

This is a great point. We did RNA-FISH with an antisense probe to CTCF. **New Supplementary Figure 1a shows Gal4-driven UAS-CTCF expression patterns in $CTCF^0$ mutants analysed in Figs. 1c-1d.** The result is that each Gal4 driver is active in largely non-overlapping cells. UAS-CTCF mRNA expression patterns resemble those of mCherry protein produced from a UAS-mCherry reporter driven by the same Gal4 drivers (as expected).

In contrast, we could not detect CTCF expressed at endogenous levels either by RNA-FISH or immunofluorescence. This could be because CTCF is expressed at low levels ($\sim 1'000$ CTCF protein molecules per nucleus in embryos according to Bonnet *et al.*, 2019 *Developmental Cell*) and to fragmentation of the CTCF RNA-FISH probe (fragmentation is critical for the probe to penetrate fixed larval brains, but can also adversely affect RNA-FISH probes).

Point 3

There is a discrepancy that needs to be clarified in Fig. 1a, 1b and Suppl. Movie 1. Some $CTCF^{KO}$ mutants (60%) hatch into adults with spasmodic movements suggesting a neurological phenotype that might be the cause of their short lifespan (Figs. 1a, 1b, Supplementary Movie 1) in contrast to “ $CTCF^0$ mutants never hatch from the pupal case (Fig. 1c)”.

Response to point 3

We apologize if our writing was unclear. As described in the original manuscript, $CTCF^{KO}$ animals have a knock-out of the *CTCF* gene but these animals still have maternally supplied CTCF mRNA and protein that partially rescues their early development. $CTCF^0$ mutants are $CTCF^{KO}$ mutants that derive from mothers lacking CTCF in their germlines, and their phenotype and lethality are more severe.

Point 4

On Fig. 2C it could be useful to visualize together the insulator scores of wt and KOs and to incorporate a heatmap outlining the differences between them.

Response to point 4

We note that original Fig. 2c already had a “delta physical insulation score” track (Fig. 2c, bottom) to show insulation score differences between WT and *CTCF⁰* mutant Hi-C maps. Above the original delta physical insulation score track, we now additionally label differential contact domain boundaries called by TopDom in only one genotype but not the other.

Point 5

*Another inconsistency is found between a conclusion stated in the text and the title of the manuscript: “These results strongly suggest that CTCF mediates formation of physical boundaries” versus the title: “CTCF loss has limited effect on genome architecture in *Drosophila* despite critical regulatory functions”.*

Response to point 5

As suggested by the reviewer, we modified the title to avoid the risk of an apparent discrepancy. It is now: “CTCF loss has a limited effect on **global** genome architecture in *Drosophila* despite critical regulatory functions”. We do not see a discrepancy between our findings that CTCF mediates formation of a small fraction of all contact domain boundaries and that CTCF is nevertheless critical for the expression patterns of certain genes and for central nervous system function.

Point 6

It is necessary to analyze the CTCF distribution and contribution to CTCF-dependent and independent boundaries in the co-existence of the so-called “architectural proteins” like BEAF-32, Cp190, Su(Hw), among others. This is relevant since it may exist additional sub-classes of boundaries associates to CTCF and other TFs and/or even co-factors.

Response to point 6

As requested by the reviewer, we **analysed the distribution of Cp190 relative to contact domain boundaries in larval central nervous systems** and found that:

- 1. Unlike CTCF, Cp190 binding is enriched at contact domain boundaries genome-wide (new Fig. 5c, new Supplementary Figures 5d and 5e).**
- 2. Outside of CTCF peaks, Cp190-occupied domain boundaries were often proximal to gene transcription start sites, suggesting a different mode of Cp190 recruitment to these loci (new Fig. 5c).**

We did not systematically profile binding of other insulator-binding proteins (IBPs) like BEAF-32 and su(Hw) that do not directly interact with CTCF (Supplementary Figure 5a). Descriptive studies of combinatorial IBP binding at contact domain boundaries defined in high-resolution Hi-C studies in *Drosophila* tissue culture cells were published (Cubéñas-Potts *et al.*, 2017; Ramírez *et al.*, 2018; Wang *et al.*, 2018). **We now summarize the view of how CTCF-independent domain boundaries may form in the Discussion:**

“In comparison to vertebrates, the principles of genome folding into contact domains in *Drosophila* are less clear. On the one hand, the majority of fly contact domains were proposed to form by compartmentalization of domains with different transcriptional states or because actively transcribed genes cluster, with little contribution from architectural proteins acting independently of transcription (Rowley *et al.*, 2017, 2019). On the other hand, analyses of enriched transcription factor motifs at domain boundaries defined at high-resolution revealed that 77% of them were enriched in core promoter motifs

(and called “promoter boundaries”) and the remaining 23% were enriched in motifs of insulator-binding proteins like CTCF, su(Hw) and Ibf1 (and called “non-promoter boundaries”) (Ramírez et al., 2018). This suggested that architectural proteins may form some domain boundaries. By completely ablating CTCF *in vivo*, we definitively show that CTCF contributes to the formation of a small fraction (below 10%) of domain boundaries in *Drosophila* (Fig. 2).”

Point 7

In line with the previous point, it is important to try to determine why some CTCF sites seem to be more prominent in comparison to others, including the author’s proposition that CTCF can reinforce boundaries redundantly. Is it relevant to discuss here which are the alternative CTCF partners and mechanisms of action ?

Response to point 7

We thank the reviewer for their insightful comment. As requested by the reviewer, we took a closer look at CTCF-occupied contact domain boundaries that are lost in *CTCF⁰* mutants (now defined as “strictly CTCF-dependent boundaries” in the manuscript) and those that are partially retained (now defined as “partially CTCF-dependent boundaries”). **New Figure 5 contrasts strictly versus partially CTCF-dependent boundaries in terms of (1) how strongly occupied they are by CTCF in WT, and (2) how much residual Cp190 binding is detected at the former CTCF peak in *CTCF⁰* mutants. We found that:**

1. In *CTCF⁰* mutants, Cp190 was frequently fully lost from former high occupancy CTCF peaks. In contrast, Cp190 was reduced but frequently partially retained at former lower occupancy CTCF peaks (new Figs. 5b and 5d). This is a more nuanced view than our previous conclusion that Cp190 was globally reduced at former CTCF peaks in *CTCF⁰* mutants.
2. Strictly CTCF-dependent boundaries were more frequently highly occupied by CTCF in WT compared to partially CTCF-dependent boundaries.
3. In *CTCF⁰* mutants, residual Cp190 binding at former CTCF-occupied boundaries was strongly associated with boundary retention (new Figs. 5d-5f). 80% of strictly CTCF-dependent boundaries lacked a residual Cp190 peak, and 79% of residual Cp190 peaks were associated with a residual boundary in *CTCF⁰* mutants (new Fig. 5e).

We conclude that either Cp190 itself or associated factors may redundantly contribute to the formation of physical boundaries independently of CTCF, and may synergize with CTCF at partially CTCF-dependent boundaries.

Point 8

CTCF distribution is explained taking advantage of a “forced” interaction with cohesin obtaining the same results as the one already published, in which cohesin interaction is not critical in fly. How this results contribute to set of results presented in the manuscript ?

Response to point 8

Direct interaction between recombinantly purified fly CTCF N-terminus and a cohesin subcomplex shown in Fig. 2g was observed using similar protein concentrations and buffer conditions used in Li *et al*, 2020. This interaction was somewhat surprising given the profound divergence between fly and human CTCF N-termini (Supplementary Figure 2f). Supplementary Figure 5a now also shows that traces of cohesin specifically co-purified with CTCF from *Drosophila* nuclear extracts analysed by mass spectrometry, reminiscent of transient interactions between CTCF and cohesin in human cells. **We more explicitly speculate about the relevance of the interaction between CTCF and cohesin in the revised Discussion: “Whether *Drosophila* CTCF, like its mammalian counterpart, forms contact domain boundaries in concert with loop-extruding cohesin remains unclear because of discrepancies between flies and mammals. (1) In mammalian Hi-C maps, CTCF sites at both anchors of an extruded loop often engage in high frequency**

contacts ⁴ that are not seen in *Drosophila* (Fig. 2c) ². (2) CTCF and cohesin colocalize genome-wide in mammals ^{49,50}, but cohesin does not colocalize specifically with CTCF in *Drosophila* ^{13,17}. (3) Mammalian CTCF-dependent boundaries are directional ^{4,5,51}; but CTCF-dependent boundaries lack clear directionality in flies (Supplementary Figure 2g) ². (4) Several regions in its N-terminus contribute to efficient loop formation by CTCF in mammals ^{10,39}, while we find that amino acid residues directly interacting with cohesin *in vitro* are conserved within an otherwise largely diverged fly CTCF N-terminus (Fig. 2g). Our finding that CTCF is required to form a small fraction of fly contact domain boundaries without strong anchor contacts could correspond to what one would nevertheless expect, given the ~10 times less frequent CTCF binding sites we report in flies (Fig. 2) relative to humans ⁴⁹, and probable differences in how fly CTCF interacts with extruding cohesin. Previous *in silico* simulations ⁶ and experiments affecting loop-extrusion processivity across CTCF-dependent boundaries in human cells ^{7,9,10} described contact domains with differing corner interaction strengths which could be similar to domains observed in flies. “

Point 9

To some extent it is not surprising that CTCF affects nearby genes. This is because the location genome-wide of CTCF in flies has shown significant proximity to TSSs. Author’s address this point but it would be relevant to analyze this aspect in more detail. Mutations can be introduced in CTCF sites located distal or proximal to a selected CNS genes and determine their differential effects, if there is any ?

Response to point 9

The suggestion to use CRISPR-Cas9 to precisely delete various CTCF sites and determine their effects on local gene expression is interesting but unfortunately goes beyond the scope of this manuscript.

Point 10

Among the 400 genes with expression changes, a compartment analysis may help to better correlate the gene expression changes and modifications in the spatial chromatin configuration.

Response to point 10

Great point. We now call A/B compartments at 2 kb resolution from our WT and *CTCF*⁰ Hi-C contact maps. **Eigenvector values tracks are now shown in all Hi-C screenshots in the manuscript. New Supplementary Figure 3d plots eigenvector values for all 2 kb bins in WT and *CTCF*⁰ mutant CNSs.** The result is that large changes in eigenvector values and A/B compartment switching between WT and *CTCF*⁰ mutants are very rare. This is also true for bins overlapping TSSs of differentially expressed genes with increased or decreased expression in *CTCF*⁰ mutants. We describe this in the Results: **“Few (8%) DE genes were located in different A/B compartments in *CTCF*⁰ mutants relative to WT, indicating that differential gene expression mostly occurred without large changes in higher-order spatial chromatin configuration (Supplementary Figure 3d, Supplementary Data 5).”**

Point 11a

In the enhancer-blocking series of experiments it would be relevant to define or select some of the CTCF sites based on their differential locations in the genomes (in proximity or distal to TSSs) and test them using the same assay.

Response to point 11a

Great point. **We annotated the locations of CTCF peaks tested in the reporter assay in Fig. 4c relative to nearby genes. Revised Supplementary Figure 4c shows that the tested peaks are located in various**

positions relative to nearby genes: 2 peaks overlap transcription start sites (TSSs), 2 peaks overlap transcription termination sites (TTSs), 3 peaks are in introns, 4 peaks are located at various distances (96-1,412 bp) upstream of the TSS of the nearest gene, and 3 peaks are located at various distances (2,151-10,000 bp) downstream of the TTS of the nearest gene. Classifying CTCF peaks according to proximity to a TSS, 5/14 tested peaks are within ± 500 bp of a TSS (TSS-proximal) and 9/14 are further away (TSS-distal). Interestingly, there is no apparent correlation between insulator strength and the proximity of CTCF peaks to nearby gene TSSs, and that even CTCF peaks overlapping TSSs (B, H) can function as insulators in our reporter. These findings are now summarized in the Results section.

Point 11b

In addition and using an average sequence of more or less 360 bp it could be interesting to see if there are other adjacent binding sites for additional transcription factors, in particular, architectural proteins or even novel factors or co-factors.

Response to point 11b

To address the reviewer's comment, **we re-mapped and visualized published ChIP-seq datasets generated for various insulator-binding proteins (IBPs) and cohesin in S2 cells.** New Supplementary Figure 4d shows that IBPs including Cp190, Ibf1, Ibf2, su(Hw), mod(mdg4), BEAF-32, ZIPIC and Pita appear to co-bind with CTCF on some tested fragments, but combinatorial IBP binding is not clearly predictive of insulator strength measured in the reporter assay in Fig. 4c. For example, fragment N has higher CTCF occupancy than fragment G and appears more strongly co-occupied by Ibf1, Ibf2 and BEAF-32; but fragment N is a weaker insulator than fragment G. Also, IBP occupancy on fragments A and B (weaker insulators) or C and D (stronger insulators) is similar despite their different insulator strengths measured in Fig. 4c. On the other hand, fragment F is a strong insulator and is strongly co-occupied by su(Hw) and mod(mdg4) which might boost its insulator activity (Fig. 4c). These findings are now summarized in the Results section: **"11 out of 14 tested CTCF peaks selectively reduced EGFP intensities to various degrees that globally scaled with CTCF ChIP-seq occupancy measured in S2 cells (Ong et al., 2013) (Fig. 4c) and that appeared independent of the endogenous locations of CTCF peaks relative to their nearest genes (Supplementary Figure 4c) and combinatorial co-binding with other fly insulator-binding proteins on the cloned fragments (Supplementary Figure 4d)."**

Point 12

It is not surprising to see that CP190 and CTCF are partners since it has already been demonstrated their co-localization. Therefore, authors should exploit deeply the genes that are commonly altered when those two factors are mutated and if they generate a particular phenotype in the fly.

Response to point 12

As requested by the reviewer, **we generated additional RNA-FISH probes to visualize spatial expression patterns of genes located close to a CTCF and Cp190 co-bound peak, that we detected as differentially expressed in both *CTCF⁰* and *Cp190^{KO}* larval central nervous systems by RNA-seq. We screened *CTCF⁰* and *Cp190⁰* embryos to stringently test whether the genes were differentially expressed in similar cell types at this early developmental timepoint. Revised Fig. 6 now shows another example of a co-regulated gene that, in contrast to *SP1029* shown originally, is expressed at decreased (instead of increased) levels in both *CTCF⁰* and *Cp190⁰* mutants relative to WT.** This supports our original conclusion that CTCF and Cp190 co-regulate a subset of target genes, but do not directly activate or repress these genes' expression. Transcription can increase or decrease in the absence of CTCF/Cp190 depending on the locus.

REVIEWER 2

Point 1:

The results obtained suggest that several architectural proteins may simultaneously contribute to the formation of a TAD boundary. In $CTCF^0$ cells some of the boundaries at former CTCF peaks were partially retained (Supplementary Fig 1a). When insulator score was analysed in transient transfection experiments (scatter plot in Fig. 4a) two groups of CTCF test fragments can be easily observed (above the spacer level): C-D-G-F, and A-M-B-H-N-E. It would appear that the dependency between the insulator strength and ChIP-seq counts is the same for both groups, but fragments from CDGF are much more strong insulators than the AMBHNE albeit demonstrate the same CTCF occupancy according the ChIP-seq. Taking all this into consideration it would be important to perform additional ChIP-seq experiments with antibodies against known Drosophila architectural proteins.

Response to point 1

As requested by the reviewer, **we re-mapped and visualized published ChIP-seq datasets generated for various insulator-binding proteins (IBPs) and cohesin in S2 cells.** New Supplementary Figure 4d shows that IBPs including Cp190, Ibf1, Ibf2, su(Hw), mod(mdg4), BEAF-32, ZIPIC and Pita appear to co-bind with CTCF on some tested fragments, but combinatorial IBP binding is not clearly predictive of insulator strength measured in the reporter assay in Fig. 4c. For example, fragment N has higher CTCF occupancy than fragment G and appears more strongly co-occupied by Ibf1, Ibf2 and BEAF-32; but fragment N is a weaker insulator than fragment G. Also, IBP occupancy on fragments A and B (weaker insulators) or C and D (stronger insulators) is similar despite their different insulator strengths measured in Fig. 4c. On the other hand, fragment F is a strong insulator and is strongly co-occupied by su(Hw) and mod(mdg4) which might boost its insulator activity (Fig. 4c). These findings are now summarized in the Results section: **“11 out of 14 tested CTCF peaks selectively reduced EGFP intensities to various degrees that globally scaled with CTCF ChIP-seq occupancy measured in S2 cells (Ong et al., 2013) (Fig. 4c) and that appeared independent of the endogenous locations of CTCF peaks relative to their nearest genes (Supplementary Figure 4c) and combinatorial co-binding with other fly insulator-binding proteins on the cloned fragments (Supplementary Figure 4d).”**

Point 2:

To clarify the cooperation between CTCF and Cp190 it will be important to perform Hi-C analysis on cells from Cp190-KO animals.

Response to point 2

We agree that performing Hi-C on Cp190 mutants will be important to understand how Cp190 contributes to CTCF function at the molecular level. This experiment must be performed using a different system can therefore not be done to simply extend the present study. To obtain unambiguous results, Hi-C must be performed on Cp190⁰ mutants that completely lack both maternal and zygotic Cp190. Cp190⁰ mutants arrest development at the end of embryogenesis, hence Hi-C must be performed in WT, CTCF⁰ and Cp190⁰ mutant embryos instead of larval brains; and matching ChIP-seq datasets should be re-generated at this new developmental stage.

Point 3:

Related to the ability of dCTCF to insulate neighbouring loci: it would be insightful to compare the physical insulation score profile around CTCF peaks and sites of other "architectural" proteins such as BEAF-32, Su(Hw) etc. in WT flies.

Response to point 3

As suggested by the reviewer, we compared the physical insulation score around CTCF peaks and Cp190 peaks in larval central nervous systems and found that **unlike CTCF, Cp190 binding is enriched at contact domain boundaries genome-wide (new Fig. 5c, new Supplementary Figures 5d and 5e)**. This is consistent with previous reports in tissue culture cells (for example: Ramírez *et al.*, 2018; Wang *et al.*, 2018).

We did not systematically profile binding of other insulator-binding proteins (IBPs) like BEAF-32 and su(Hw) that do not directly interact with CTCF (Supplementary Figure 5a). Descriptive studies of combinatorial IBP binding at contact domain boundaries defined in high-resolution Hi-C studies in *Drosophila* tissue culture cells were published (Cubebñas-Potts *et al.*, 2017; Ramírez *et al.*, 2018; Wang *et al.*, 2018). **We now summarize the view of how CTCF-independent domain boundaries may form in the Discussion (“Relaxed requirement of CTCF for *Drosophila* genome architecture”):**

“In comparison to vertebrates, the principles of genome folding into contact domains in *Drosophila* are less clear. On the one hand, the majority of fly contact domains were proposed to form by compartmentalization of domains with different transcriptional states or because actively transcribed genes cluster, with little contribution from architectural proteins acting independently of transcription (Rowley *et al.*, 2017, 2019). On the other hand, analyses of enriched transcription factor motifs at domain boundaries defined at high-resolution revealed that 77% of them were enriched in core promoter motifs (and called “promoter boundaries”) and the remaining 23% were enriched in motifs of insulator-binding proteins like CTCF, su(Hw) and Ibf1 (and called “non-promoter boundaries”) (Ramírez *et al.*, 2018). This suggested that architectural proteins may form some domain boundaries. By completely ablating CTCF *in vivo*, we definitively show that CTCF contributes to the formation of a small fraction (below 10%) of domain boundaries in *Drosophila* (Fig. 2).”

Point 4:

I suppose the authors should describe their findings on CD borders in more details in the main text: how many borders were identified in WT and in the CTCF-0, and how many borders lost in the CTCF-0 were bound with CTCF in WT.

Response to point 4

We previously faced the following challenges in counting contact domain boundaries: calling of contact domain boundary by TopDom was either not sensitive enough (leading to an underestimation of CD boundaries at CTCF peaks) or too sensitive (leading to many ‘noisy’ boundaries). To provide the CD boundary counts requested by the reviewer, we optimized CD boundary calling and revised the Methods section: (1) TopDom window sizes were adapted from 10, 20, 40, 80, 120 and 240 kb (used previously) to 20, 40, 80 and 160 kb. Eliminating the smaller window size leads to less noisy boundary calling. (2) Groups of consecutive TAD boundaries were merged to avoid mismatches between genotypes caused by small fluctuations of CD boundary positions (obtained with different window sizes or in different genotypes) caused by random noise. Concretely, groups of consecutive (i.e. at distance ‘binsize’) boundaries (found in any of the two genotypes) were replaced by the boundary having the lowest insulation score (average of both genotypes for common boundaries or of one genotype for boundaries unique to one genotype).

New Supplementary Table 2 indicates how many CD boundaries were identified in WT and/or in *CTCF*⁰, and how many of these boundaries were bound by CTCF in WT. New Fig. 2d shows the enrichment of CTCF peaks around all CD boundaries identified in any genotype (whereas previously we only plotted CD boundaries weakened in *CTCF*⁰), ranked by physical insulation score differences measured in *CTCF*⁰ minus WT Hi-C maps (weaker boundaries in *CTCF*⁰ are at the top). Boundaries were color-coded as present in both WT and *CTCF*⁰ (blue), present only in WT (red) or present only in *CTCF*⁰ (green) to globally visualize their relative frequencies. The results are described in the revised Results section: “WT and *CTCF*⁰ Hi-C

maps were globally similar, and most (84%) domain boundaries were detected in both WT and *CTCF⁰* mutants. Nevertheless, specific contact domains were visibly less physically insulated from the neighboring domain in *CTCF⁰* mutants (Fig. 2c, Supplementary Fig. 2c, Supplementary Table 3). Clearly disrupted domain boundaries in *CTCF⁰* mutants frequently occurred at former CTCF peaks (Fig. 2d). Of 136 strongly affected domain boundaries that were lost in *CTCF⁰* mutants, 89 (65%) were at former CTCF peaks (Supplementary Table 2)."

The updated CD boundary calls have led to revised versions of the following figures:

- Updated physical insulation score tracks (the window sizes changed and a new average physical insulation score was obtained) and CD boundaries are now shown in all Hi-C screenshots.
- Fig. 2a: 33% (formerly 44%) of CTCF peaks are within ± 1 kb of a CD boundary, representing a 6-fold (formerly 5-fold) enrichment over random regions.
- Fig. 2b: 8% (formerly 6%) of CD boundaries are within ± 1 kb of a CTCF peak, representing a 6-fold (formerly 5-fold) enrichment over random regions.
- Fig. 2d: all new CD boundaries called in any genotype are shown.
- Fig. 2e: new CD insulation scores (calculated with the new TopDom window sizes) are now plotted.
- Fig. 2f: new average physical insulation scores (calculated with the new TopDom window sizes) are now plotted.

Point 5:

*The authors state: "The vast majority of strongly affected domain boundaries in *CTCF⁰* mutants occurred at former CTCF peaks (Fig. 2d)". The authors should show a completely marked vertical axis in Fig. 2d, because in the current version it is impossible to evaluate the actual range of insulation defects at CTCF-bound CD borders "strongly affected" in *CTCF*-deficient flies.*

Response to point 5

As requested by the reviewer, **Fig. 2d now has a completely marked vertical axis.** To further highlight what we consider to be clearly disrupted boundaries in *CTCF⁰*, **boundaries with a delta physical insulation score in *CTCF⁰* minus WT Hi-C maps > 0.1 are bracketed.**

Point 6:

*The authors propose that "... *CTCF* is required to form physical boundaries with strengths generally proportional to its occupancy on DNA".*

*However, in my opinion, this is a broad statement. What do we actually see in the results? (i) not all but only a minor fraction (what proportion exactly?) of *CTCF*-bound CD borders are lost in the absence of *CTCF*;*

*(ii) Figure 2d shows that the mostly affected CD borders are bound with *CTCF*, but does not allow one to accurately estimate the dependence between the IS defect value and *CTCF* occupancy at the border.*

*Thus, the authors should indicate the number of lost (or significantly weakened) CD borders in the *CTCF*-0, and provide a quantitative analysis of the relationships between *CTCF* occupancy and the degree of the IS defects at the CD borders.*

Response to point 6 (i)

As requested by the reviewer, the number of *CTCF*-bound CD boundaries that are lost in *CTCF⁰* mutants are now provided in **Supplementary Table 2** and now described in the Results section: **"Of 350 contact domain boundaries bound by *CTCF* in WT, only 131 (37%) were fully lost in *CTCF⁰* (Supplementary Table 2)."**

Response to point 6 (ii)

As requested by the reviewer, we now **plot for every CTCF peak, the physical insulation score differences observed in *CTCF⁰* mutant relative to WT Hi-C maps as a function of CTCF occupancy measured by CHIP-seq. New Supplementary Figure 2d** supports our original conclusion that CTCF is required to form physical boundaries with strengths generally proportional to its occupancy on DNA.

Point 7:

*The authors conclude that "...the pervasive weakening of physical boundaries observed at former CTCF peaks in *CTCF⁰* mutants (Fig. 2f) is not a mere consequence of altered transcription". However, this statement is based on visual inspection of limited number of gene loci. Meanwhile, the interplay between transcription and CTCF-driven chromatin 3D organization is far from the complete description, and the unique model system presented in this work provides a good opportunity to get some novel insights. I suggest the authors could annotate spatial contacts between promoters of DE genes and nearby enhancers, and to investigate whether expression deregulation is caused by formation or disruption of loops with CTCF-bound regulatory elements.*

Response to point 7

To address the reviewer's comment, **new Figure 5d shows the distribution of differentially expressed (DE) genes in *CTCF⁰* mutants, around all 358 CTCF-occupied contact domain boundaries.** This highlights the fact that DE genes are only present at a fraction of CTCF-dependent boundaries, and the severity of boundary disruption in *CTCF⁰* mutants does not appear linked to the presence/absence of a detectable DE gene nearby. We also now called A/B compartments at 2 kb resolution from our WT and *CTCF⁰* Hi-C contact maps as described in the revised Methods section. **New Supplementary Figure 3d plots eigenvector values for all 2 kb bins in WT and *CTCF⁰* mutant CNSs. The result is that large changes in eigenvector values and A/B compartment switching between WT and *CTCF⁰* mutants are very rare, even for bins overlapping TSSs of DE genes.** This point is now included in the revised Results section: "**Few (8%) DE genes were located in different A/B compartments in *CTCF⁰* mutants relative to WT, indicating that differential gene expression mostly occurred without large changes in higher-order spatial chromatin configuration (Supplementary Figure 3d, Supplementary Data 5).**"

Addressing whether gain/loss of 3D contacts between DE gene promoters and annotated enhancers explains gene misexpression patterns in *CTCF⁰* mutants is exciting but daunting. Enhancers have not been annotated in third instar larval brains, though single-cell ATAC-seq studies are underway in some laboratories. This exercise would not answer the question of whether differential expression is a cause or a consequence of altered 3D contacts.

REVIEWER 3

Point 1:

Authors should include a Supplementary Table with QC results of Hi-C libraries, such as those obtained using Juicer to process the data. In particular, information on intra- versus inter-chromosome interactions and contacts <20 kb and >20 kb would help the reader judge the quality of the Hi-C data.

Response to point 1

The information requested by the reviewer is provided in new Supplementary Table 1 and referenced in the Results: “Hi-C maps consisting of 200 million reads per genotype were obtained by combining the correlated biological replicates (see Methods, **Supplementary Table 1**.” Note that as described in the original Methods, Juicer was not used to process the data.

Point 2:

825 CTCF peaks seems like a low number. Was the CTCF antibody described previously and its specificity tested? Please include either a description of previously published information or details on the characterization of the antibody

Response to point 2

The reviewer rightly points out that we report much fewer CTCF peaks than previous studies (ranging from 2'000 peaks reported by *Nègre et al.*, 2010 to 6'000 CTCF peaks reported by *Ong et al.*, 2013).

A rabbit polyclonal CTCF antibody was generated for this study and was described in the original “Antibodies” paragraph in the Methods section. That this antibody specifically recognizes CTCF is demonstrated by the global loss of WT ChIP-seq signals in *CTCF⁰* (Figs. 2,3,5,6 and Supplementary Figures 2,3,5,6). **New Supplementary Figure 2a now presents anti-CTCF Western blots on whole-cell extracts prepared from WT, *CTCF⁰* and CTCF-overexpressing larval central nervous systems and shows that the antibody specifically recognizes full-length CTCF.** The Results section was revised: “In parallel, CTCF binding sites were mapped in larval CNSs by chromatin immunoprecipitation sequencing (ChIP-seq) with a polyclonal antibody **specifically recognizing CTCF (Supplementary Figure 2a)** in WT and *CTCF⁰* animals as control.”

Rather than depending on use of an independent CTCF antibody, the discrepancy between the number of CTCF peaks reported by us and others likely arises from our peak calling procedure (described in the original Methods). We stringently called initial peaks using csaw v1.16.1 (Lun *et al.*, 2015) with FDR 0.01, but most crucially, we performed control ChIP-seq experiments in mutants completely lacking CTCF for the first time. Our final CTCF peaks were defined as enriched in WT relative to *CTCF⁰* mutants, as described in the original Results (“CTCF peaks were defined as enriched in WT relative to *CTCF⁰* CNSs”) and in the original Methods (“Genuine CTCF and Cp190 peaks were identified by differential analyses of ChIP-seq signals in WT versus *CTCF⁰* and WT versus *Cp190^{KO}*, respectively, as being lower in the mutant sample controls relative to WT”). We clarify that 821 CTCF peaks were obtained by differential analysis between WT versus *CTCF⁰* ChIP data. **These peaks are now provided in new Supplementary Data 1. 825 CTCF peaks were obtained after splitting a few of these peaks to allow three-way comparisons shown in Fig. 5a. This is now explained in the Methods: “To count overlaps between CTCF and Cp190 peaks in three-way comparisons shown in Fig. 5a, some CTCF and Cp190 peaks were split into 2 or 3 sub-regions. Specifically, 821 WT CTCF peaks were split into 825 peaks, 6,473 WT Cp190 peaks were split into 6,477 peaks, and 1,045 differentially bound Cp190 regions with lower occupancy in *CTCF⁰* relative to WT were split into 1,080 peaks.”**

To assess whether our CTCF peaks were also found in other studies, we overlapped our CTCF peaks with the list of CTCF peaks published by Ong *et al.*, 2013 in S2 cells (GEO accession GSE41354). (In contrast to Ong *et al.*, other studies did not provide a list of called CTCF peaks.) 805/821 (98%) of our peaks were within

20 bp of a CTCF peak from Ong *et al.*, 2013, but only 793/6'189 peaks (13%) of CTCF peaks reported by Ong *et al.*, 2013 were among our peaks. Recent publications raised concerns that several reported *Drosophila* CTCF ChIP peaks could be unspecific because they lacked a clear CTCF motif in the underlying DNA sequence (Jain *et al.*, 2015; Ramírez *et al.*, 2018). We determined the proportion of CTCF peaks overlapping at least one CTCF consensus motif (identified in the reference genome using FIMO and the JASPAR CTCF insect motif MA0531.1): 80% of our CTCF peaks and 22% of peaks from Ong *et al.*, 2013 had a CTCF motif within a 1 kb window around the center of a CTCF peak. Finally, we assessed the enrichment of CTCF motifs within the 821 top-scoring peaks reported by Ong *et al.*, 2013 after removing the 805 common peaks that were in both of our lists: 45% of the next 821 top-scoring peaks from Ong *et al.* had a CTCF motif within a 1 kb window around the center of a CTCF peak. We conclude that our stringent definition of CTCF peaks may have led to false-negatives. Some differential peaks between our dataset and that of Ong *et al.* may alternatively be due to cell-type specific differences in CTCF binding between S2 cells and larval central nervous systems. Importantly, however, we appear to have significantly less false-positive peaks as CTCF motifs are 4-fold enriched under our peaks compared to peaks from Ong *et al.*, 2013. **The results of the comparison between our peaks and those of Ong *et al.* are now presented in a new Supplementary Figure 2b, and discussed in the revised Results section: "821 CTCF peaks were defined as enriched in WT relative to *CTCF*⁰ CNSs, of which 80% overlapped a CTCF consensus motif (Supplementary Figure 2b, Supplementary Data 1)."**

The fact that we find only 821 CTCF peaks is central to how we interpret our results. *Drosophila* contact domain boundaries outnumber CTCF peaks (Fig. 2), and most contact domain boundaries are (as expected) CTCF-independent.

Point 3:

"These results strongly suggest that CTCF mediates formation of physical boundaries". Previous work has shown that, what appear to be boundaries between domains observed using low resolution Hi-C, of the same order of contacts employed here, are in fact small domains. It is possible that what appears to be physical insulation is caused by compartmental interactions between small active compartments flanking large compartments containing no genes or silenced genes. One possibility to distinguish between the two possibilities would be to compare the Hi-C data described here with that obtained in cultured cells by, for example, Eagen et al 2017 or Rowley et al 2018, which contain around 1 billion valid contacts. This comparison could be performed in regions of the genome containing the same transcribed genes and CTCF peaks. This issue is conceptually important i.e. does Drosophila CTCF mediate compartmental interactions between active domains or does it create boundaries by some unknown mechanism that does not involve the formation of loops like in mammals?

Response to point 3

To address the reviewer's concern that contact domain (CD) boundaries called in our study could correspond to small (active) contact domains that are only resolved in higher-resolution Hi-C contact maps (analysed at 500 bp resolution instead of 2 kb used in our study), we compared our CD boundary calls to CD coordinates published by Eagen *et al.*, 2017 and Ramírez *et al.*, 2018 in Kc167 tissue culture cells. [Note that Ramírez *et al.*, 2018 analysed Hi-C data from Cubeñas-Potts 2017 similar to Rowley *et al.*, 2017, and provided a list of CD coordinates as a supplementary file.] How many of our CD boundaries could be small domains identified by Eagen and Ramírez? To answer this, we counted how many small (≤ 4 kb) contact domains identified in those published studies were close (within 2kb) to one of our CD boundaries. We could have potentially mis-called such small domains close to our CD boundaries as a CD boundary. The result is that Eagen did not identify contact domains smaller than 6 kb. Only 31 of our CD boundaries were within 2 kb of a ≤ 4 kb CD identified by Ramírez. Thus, very few (31/3'970, or $<1\%$) of our CD boundaries may correspond to small active domains defined by Ramírez. We next asked: How many of our WT-specific boundaries (i.e. that disappear in *CTCF*⁰ Hi-C maps) could correspond to small active domains? The result is

that very few (4/567, or <1%) of our WT-specific CD boundaries were within 2 kb of a ≤ 4 kb CD identified by Ramírez.

This point is now included in the Results section: “Very few (<1%) of the boundaries defined in this study potentially correspond to small contact domains defined in even higher resolution Hi-C studies (see Methods)”. A new paragraph in the Methods section describes these analyses: “Comparing contact domain boundary calls with higher resolution Hi-C studies”.

To enable the reader to visually compare the positions of our CD boundaries to those called by Ramírez *et al.*, these are now displayed together in all Hi-C screenshots shown throughout the manuscript. Note that for simplicity, we did not additionally display CD boundaries called by Eagen *et al.* because several of these boundaries flanked inter-TAD regions. In other words, some CD boundaries from Eagen *et al.* are close and could mistakenly be interpreted as delimiting small contact domains between them. For brevity and to not distract the reader from our results, we refrain from going into these details in our figure legends. Overall, the result is that our CD boundaries globally align with those of higher resolution studies, although differences in cell types analyzed (embryonic tissue culture cells versus larval central nervous systems) and CD boundary calling strategies prevent rigorous comparisons between datasets.

The CTCF-dependent boundaries that we define therefore do not appear to be small domains. Although we performed our analyses at 2 kb instead of 500 bp resolution, our Hi-C studies are not comparable to low-resolution studies that mis-labeled small active domains as domain boundaries using Hi-C maps at ~ 20 kb resolution.

Point 4:

Figure 2g. Although the results in this figure look very clean and convincing, they are difficult to rationalize. The YDF residues of human CTCF interact with a pocket formed by SA2 and SSC1, and several residues of SSC1, which is not present in the experiments described in Figure 4g, are necessary for the interaction. Authors should comment on this.

Response to point 4

Fig. 2g tests whether the direct interaction reported between human CTCF and human SA2 and SCC1 by Li *et al.*, 2020 is conserved between homologous fly CTCF, SA and Vtd protein fragments. Li *et al.* showed that human CTCF N-terminal peptides comprising the YDF residues directly interact (and co-crystalize) with a complex formed by human SA2 residues 80-1060 and human SCC1 residues 281-420. Alignments of fly CTCF/human CTCF (shown in original Supplementary Fig. 2d), fly SA/human SA2 and fly Vtd/human SCC1 proteins (shown below) demonstrates that homologous fragments were tested in Li *et al.* and this study, except that we used the entire CTCF N-terminus instead of smaller peptides used by Li *et al.* (because we had already purified CTCF N-terminus for the proteomics experiments described in Supplementary Figure 5a). In the fly SA/human SA2 and fly Vtd/human SCC1 protein alignments shown below, the fragments tested by Li *et al.* and by us are highlighted in red. Amino acid residues in SA2 and SCC1 that directly contact CTCF in the published crystal structure (see Fig. 1 of Li *et al.*) are boxed in blue and green, respectively. This shows that no CTCF-interacting residues in SCC1 are missing from the Vtd protein construct tested in Fig. 2g.

SA2/SA protein alignment (human construct: 80-1060; fly construct: 102-1085)

SCC1/Vtd protein alignment (human construct: 281-420; fly construct: 273-458)

Point 5:

Figure 3. It has been shown that some CTCF sites in Drosophila co-localize with SuHw whereas others do not. Since SuHw has been shown to repress transcription, at least under some circumstances, it would be interesting to examine whether there is a correlation between up or down-regulation of transcription in CTCF mutants and the presence or absence of SuHw at a subset of sites. Since SuHw has a well-defined binding motif, it may be sufficient to look for the motif adjacent to CTCF sites without having to perform ChIP-seq.

Response to point 5

To address the reviewer's comment, we first identified transcription start sites (TSSs) of genes with increased (UP) or decreased (DOWN) expression in *CTCF⁰* mutant larval central nervous systems that had a CTCF peak within ± 2 kb (i.e. a less stringent cut-off than the ± 1 kb cut-off used in Fig. 3f). The result was that 17% of UP genes and 18% of DOWN genes have a CTCF peak within ± 2 kb. In a second step, we searched for the presence of at least one *su(Hw)* motif (identified in the reference genome using FIMO and the JASPAR *su(Hw)* motif MA0533.1) present within ± 500 bp from the center of the CTCF peaks identified in the first step. The result was that 2% of UP genes and 4% of DOWN genes had at least one CTCF peak within ± 2 kb that was itself within ± 500 bp (from the CTCF peak center) to at least one *su(Hw)* motif. We conclude that we do not find evidence that *su(Hw)* binding in the vicinity of CTCF peaks is either strongly or differentially enriched around genes with increased or decreased expression in *CTCF⁰* mutants. We note that *su(Hw)* protein was not enriched in our CTCF pull-downs analyzed by mass spectrometry (Supplementary Figure 5a). It remains, however, possible that CTCF and *su(Hw)* indirectly functionally synergize at a subset of loci.

Point 6a:

Figure 4. Authors should explain the hypothesis for how the test fragments containing CTCF sites would affect enhancer-promoter interactions in the experiments described in Figure 4. The plasmids do not integrate in the chromosomes and are presumed to be circular. Therefore, in principle, the enhancer could contact the promoter in either direction around the plasmid. Perhaps having a second site to insert the test fragments would have made the results easier to interpret. Authors should discuss these possible shortcomings in the text.

Response to point 6

We thank the reviewer for raising this point. We indeed found that having a second insulator downstream of EGFP, such that EGFP is flanked on both sides by insulators, greatly boosted insulator activity. Raw scatter plots (like those shown in Supplementary Figure 4a) illustrating this point are shown below for the reviewer: (1) reporter without any insulators, (2) reporter with a single *gypsy* insulator downstream of EGFP, (3) reporter with a single *gypsy* insulator upstream of EGFP, (4) reporter with 2 *gypsy* insulators flanking EGFP. The experiments shown in this manuscript were therefore all performed in reporters containing a *gypsy* insulator in the plasmid backbone, downstream of EGFP. Even though *gypsy* is a *su(Hw)*-dependent insulator, we found that it synergized with CTCF-dependent insulators tested in Fig. 4. We omitted mention of this second insulator in the original manuscript for simplicity, to not distract the reader from the cloned insulators. But the reviewer is right - it is necessary to show the second insulator to properly interpret the results. Therefore, **the insulator reporter schematic shown in Fig. 4a was updated and now depicts the location of the *gypsy* insulator (drawn to scale). The reporter is now described in full detail in the legend of revised Fig. 4a: "A *gypsy* insulator G is present downstream of EGFP to block EGFP activation by the enhancer (which in a circular plasmid molecule is both upstream *and* downstream of EGFP) from the left."**

Point 6b:

It is probably not appropriate to use results from these experiments to speculate what CTCF does in the normal situation. The conclusion “(4) Sites bound by CTCF do not directly repress or activate transcription, but rather functionally insulate promoters and enhancers in a reporter assay in S2 cells” in the Discussion is too strong.

Response to point 6b

We agree that it is not appropriate to extrapolate CTCF’s activities in the reporter assay to what CTCF does at endogenous chromosomal loci, hence the cited conclusion from the original manuscript specified “in a reporter assay in S2 cells”. We did not remove this result from our conclusions summary because we think it is important to highlight this point since an alternative explanation for the results in Fig. 4 could have been that CTCF-bound sites in the reporter decrease EGFP expression by acting as repressors instead of enhancer-blockers.

Point 7:

“Cp190 colocalized with CTCF at nearly all (77%) CTCF peaks”. At least for me, “nearly all” would be 99%. Please let the reader decide and delete “nearly all”.

Response to point 7

As requested by the reviewer, “**most**” now replaces “nearly all”. We note that browsing the raw ChIP-seq data, we did not find clear examples in which CTCF bound independently of Cp190. ChIP experiments even with two independent antibodies against the same protein do not result in extremely high overlaps between ChIP-seq peaks in our experience, due to inherent “noise” in data generation and analysis.

Point 8:

Figure 5f. It would be interesting if the authors came up with a more sophisticated model than that shown in Figure 5f. In mammals, the “boundary” effect of CTCF on E-P interactions can be explained by cohesin extrusion, which may increase the frequency of interactions within the loop but, by stopping at CTCF convergent sites, decreases interactions between regulatory sequences inside and outside of the loop. As a consequence, the CTCF site appears to form a boundary that interferes with E-P interactions. It would be surprising if cohesin is unable to extrude in Drosophila. However, the absence of puncta at Drosophila domains “flanked” by CTCF suggest that, even if there is extrusion by cohesin, this complex does not appear to stop at CTCF sites. Results from high resolution Hi-C in Drosophila cells suggest that the “boundaries” are in fact small active domains that interact with other active domains in the A compartment. It’s interesting that CTCF and other Drosophila architectural proteins form large puncta in IF experiments (see for example Gerasimova et al 2008, where CTCF and CP190 are shown to co-localize at large dots in the nucleus). These puncta are similar to what would be considered membraneless organelles formed by LLPS. Proteins like CTCF and CP190 may work by increasing the frequency

or stability among *A* compartmental domains. It would be interesting to put all these facts together into a congruent model.

Response to point 8

To address the reviewer's comment, our revised Discussion more explicitly discuss potential mechanisms by which CTCF may form contact domain boundaries in *Drosophila*, and consider the models mentioned by the reviewer of loop-extrusion versus coalescence of CTCF bound to distal loci:

“Whether *Drosophila* CTCF, like its mammalian counterpart, forms contact domain boundaries in concert with loop-extruding cohesin remains unclear because of discrepancies between flies and mammals. (1) In mammalian Hi-C maps, CTCF sites at both anchors of an extruded loop often engage in high frequency contacts⁴ that are not seen in *Drosophila* (Fig. 2c)². (2) CTCF and cohesin colocalize genome-wide in mammals^{49,50}, but cohesin does not colocalize specifically with CTCF in *Drosophila*^{13,17}. (3) CTCF-dependent boundaries are directional in mammals^{4,5,51}; but lack clear directionality in flies (Supplementary Figure 2g)². (4) Several regions in its N-terminus contribute to efficient loop formation by CTCF in mammals^{10,39}, but only a minimal stretch of amino acid residues that directly interact with cohesin *in vitro* (Fig. 2g) are visibly conserved in fly CTCF. Our finding that CTCF is required to form a small fraction of fly contact domain boundaries without strong anchor contacts could correspond to what one would nevertheless expect, given the ~10 times less frequent CTCF binding sites we report in flies relative to humans⁴⁹, and probable differences in how fly CTCF interacts with extruding cohesin. Previous *in silico* simulations⁶ and experiments affecting loop-extrusion processivity across CTCF-dependent boundaries in human cells^{7,9,10} described contact domains with differing corner interaction strengths which could resemble domains observed in flies. We note that other studies have alternatively proposed that CTCF forms chromosomal loops through extrusion-independent mechanisms in flies, for example by binding to distal loci and coalescing⁵² or dimerizing, either directly via CTCF's N-terminus⁴⁵ or indirectly by recruiting Cp190 which dimerizes through its BTB domain⁵³.”

As suggested by the reviewer, we now present a more elaborate summary model. **New Fig. 6f summarizes all our major findings**, but refrains from speculating about mechanisms for which we lack direct evidence.

REVIEWER 4

Point 1:

“10% of DE genes had a CTCF peak within ± 1 kb of their transcriptional start site (TSS). Is that equally distributed amongst up- and down- regulated genes?”

Response to point 1

To answer the reviewer’s question, we analyzed CTCF peak enrichment separately around genes with decreased (DOWN) and increased (UP) expression in *CTCF⁰* versus WT CNSs relative to matching numbers of random genes. **New Supplementary Figure 3c shows that CTCF peaks are somewhat more enriched around DOWN genes (9-fold enrichment) than around UP genes (6-fold enrichment).** This finding is summarized in the Results section: “10% of DE genes had a CTCF peak within ± 1 kb of their transcriptional start site (TSS) (8-fold enrichment over non-DE genes) (Fig. 3f), **a result that was not very different for genes with increased versus decreased expression in *CTCF⁰* mutants (Supplementary Figure 3c).**”

New Figure 5d, lane 6 shows the distribution of differentially expressed genes (UP in red, DOWN in blue) around CTCF-occupied contact domain boundaries. CTCF peaks were not visibly enriched very close (± 100 bp) to the transcription start sites of DOWN genes as observed in mouse ESCs (Nora *et al.*, 2017).

Point 2

*“When presenting the changes in CP190 occupancy in the *CTCF⁰* mutant it would be helpful to show density heatmaps of CP190 ChIP-seq signal centered at CTCF peaks, in the WT and *CTCF⁰* mutant side by side. Ideally in the main figures, as opposed (or in addition to) the Venn diagram of figure 5b.”*

Response to point 2

As requested by the reviewer, **new Figure 5b presents heatmaps of raw Cp190 ChIP-seq signal around CTCF peaks in WT and *CTCF⁰* mutants.** We thank the reviewer for this helpful comment because it revealed an important nuance in our previous conclusion that Cp190 binding is globally reduced at former CTCF peaks in *CTCF⁰* mutants. We now observe that in *CTCF⁰* mutants, Cp190 is frequently fully lost from former high occupancy CTCF peaks, but is partially retained at former lower occupancy CTCF peaks. This result and how it relates to retention of residual boundaries in *CTCF⁰* mutants are described in new Fig.5c-g.

Point 3

*“It would be interesting if authors could speculate further about why corner peaks are not detected at CTCF-dependent contact domain boundaries, in light of the CTCF-SA-Vtd/Rad21 interaction being conserved in *Drosophila*. Do authors think it is a detection issue of the Hi-C, or something inherently different in how CTCF and cohesin cooperate in *Drosophila*? Do these findings imply that the CTCF-SA-Rad21 interaction in mammals is unlikely to be sufficient to create Hi-C peaks, and that additional (potentially mammalian-specific) actors must be involved downstream?”*

Response to point 3

As requested by the reviewer, our revised Discussion speculates further about the relevance of fly CTCF’s ability to interact with cohesin:

“Whether *Drosophila* CTCF, like its mammalian counterpart, forms contact domain boundaries in concert with loop-extruding cohesin remains unclear because of discrepancies between flies and mammals. (1) In mammalian Hi-C maps, CTCF sites at both anchors of an extruded loop often engage in high frequency contacts ⁴ that are not seen in *Drosophila* (Fig. 2c) ². (2) CTCF and cohesin colocalize genome-wide in mammals ^{49,50}, but cohesin does not colocalize specifically with CTCF in *Drosophila* ^{13,17}. (3) CTCF-dependent boundaries are directional in mammals ^{4,5,51}; but lack clear directionality in flies

(Supplementary Figure 2g) ². (4) Several regions in its N-terminus contribute to efficient loop formation by CTCF in mammals ^{10,39}, but only a minimal stretch of amino acid residues that directly interact with cohesin *in vitro* (Fig. 2g) are visibly conserved in fly CTCF. Our finding that CTCF is required to form a small fraction of fly contact domain boundaries without strong anchor contacts could correspond to what one would nevertheless expect, given the ~10 times less frequent CTCF binding sites we report in flies relative to humans ⁴⁹, and probable differences in how fly CTCF interacts with extruding cohesin. Previous *in silico* simulations ⁶ and experiments affecting loop-extrusion processivity across CTCF-dependent boundaries in human cells ^{7,9,10} described contact domains with differing corner interaction strengths which could resemble domains observed in flies.”

REVIEWERS' COMMENTS

Reviewer #1 (Remarks to the Author):

The manuscript by Gambetta MC, Lieberman-Aiden E and collaborators with the title: "CTCF loss has limited effect on genome architecture in Drosophila despite critical regulatory functions" report the effects of CTCF elimination on genome architecture in Drosophila. The manuscript has been extensively reviewed and new data incorporated.

- The unified model seems to be appropriate and helpful for the reader (Figure 6).
 - The CTCF and CP190 relationship has now been clarified not only with author's data but also with the data from the literature.
 - The levels of CTCF in the knock-out context are examined by RNA-FISH. This is a relevant control.
 - The analysis of "architectural proteins" associated differentially to distinct boundary regions contribute to alternative and complementary conclusions.
 - The CTCF-cohesin relationship is now addressed and discussed in more detail.
 - The supplementary analysis of the 400 genes with differences in gene expression is useful and expands the interpretation of the results obtained in relation to the A/B compartments.
 - Authors as suggested by this reviewer revised the enhancer-blocking sets of assays. In addition the nearby interactions (in a 360 bp range) of other "architectural proteins" has also been surveyed.
- In conclusion and with no doubt the manuscript has improved, the results are now clearly described; they are relevant and appropriate for publication in Nature Communications.

Reviewer #2 (Remarks to the Author):

The authors have properly addressed my comments or explained why they cannot perform the proposed additional experiments. Taking into consideration the importance of the main finding (dispensability of CTCF for the formation of the major part of contact chromatin domain boundaries in Drosophila) I recommend the revised version of the MS for publication in Nature Communications.

Reviewer #3 (Remarks to the Author):

The manuscript describes an analysis of 3D chromatin organization in Drosophila with an emphasis on the role of CTCF. The authors examine the effect of mutations in CTCF, find a specific effect on the CNS, and examine 3D organization and gene expression in this specific tissue. Authors find a requirement of CTCF for the maintenance of a subset of domain boundaries and on the transcription of a subset of genes. Authors then explore the role of CP190, which colocalizes with CTCF in a subset of genomic sites.

I found the manuscript very interesting and the results significant. The authors analyze the Hi-C data in a manner that makes sense and leads to biological insights, rather than following what has become standard in the field, just looking at whether TADs come or go. The observation that CTCF does not form loops like in mammals but still has a functional YDF motif that interacts with cohesin is very interesting and brings into question whether this domain is sufficient for cohesin interaction. I wonder if Drosophila has a CTCFL homolog that also has the YDF motif.

My only suggestion for the authors would be to examine the effect of CTCF sites on boundary maintenance and gene expression by looking separately at sites that are close to the promoter and sites that are not. *Drosophila* intergenic regions are very short, and to do this analysis the authors would have to consider the promoter as +/- 200 bp, more or less. My understanding is that CTCF sites in *Drosophila* have a bimodal distribution and fall into two categories, proximal and distal to the promoter. It is possible that promoter sites are directly involved in transcription whereas distal sites, which tend to co-localize with SuHw sites, may have the enhancer blocking function.

Point by point response to reviewer comments on revised manuscript

REVIEWER 1

The manuscript by Gambetta MC, Lieberman-Aiden E and collaborators with the title: “CTCF loss has limited effect on genome architecture in Drosophila despite critical regulatory functions” report the effects of CTCF elimination on genome architecture in Drosophila. The manuscript has been extensively reviewed and new data incorporated.

- The unified model seems to be appropriate and helpful for the reader (Figure 6).*
- The CTCF and CP190 relationship has now been clarified not only with author’s data but also with the data from the literature.*
- The levels of CTCF in the knock-out context are examined by RNA-FISH. This is a relevant control.*
- The analysis of “architectural proteins” associated differentially to distinct boundary regions contribute to alternative and complementary conclusions.*
- The CTCF-cohesin relationship is now addressed and discussed in more detail.*
- The supplementary analysis of the 400 genes with differences in gene expression is useful and expands the interpretation of the results obtained in relation to the A/B compartments.*
- Authors as suggested by this reviewer revised the enhancer-blocking sets of assays. In addition the nearby interactions (in a 360 bp range) of other “architectural proteins” has also been surveyed. In conclusion and with no doubt the manuscript has improved, the results are now clearly described; they are relevant and appropriate for publication in Nature Communications.*

Response to the reviewer’s comments

We thank the reviewer for their original comments and are pleased that these were adequately addressed.

REVIEWER 2

The authors have properly addressed my comments or explained why they cannot perform the proposed additional experiments. Taking into consideration the importance of the main finding (dispensability of CTCF for the formation of the major part of contact chromatin domain boundaries in Drosophila) I recommend the revised version of the MS for publication in Nature Communications.

Response to the reviewer's comments

We thank the reviewer for their original comments and are pleased that these were adequately addressed.

REVIEWER 3

*The manuscript describes an analysis of 3D chromatin organization in *Drosophila* with an emphasis on the role of CTCF. The authors examine the effect of mutations in CTCF, find a specific effect on the CNS, and examine 3D organization and gene expression in this specific tissue. Authors find a requirement of CTCF for the maintenance of a subset of domain boundaries and on the transcription of a subset of genes. Authors then explore the role of CP190, which colocalizes with CTCF in a subset of genomic sites.*

*I found the manuscript very interesting and the results significant. The authors analyze the Hi-C data in a manner that makes sense and leads to biological insights, rather than following what has become standard in the field, just looking at whether TADs come or go. The observation that CTCF does not form loops like in mammals but still has a functional YDF motif that interacts with cohesin is very interesting and brings into question whether this domain is sufficient for cohesin interaction. I wonder if *Drosophila* has a CTCFL homolog that also has the YDF motif.*

*My only suggestion for the authors would be to examine the effect of CTCF sites on boundary maintenance and gene expression by looking separately at sites that are close to the promoter and sites that are not. *Drosophila* intergenic regions are very short, and to do this analysis the authors would have to consider the promoter as +/- 200 bp, more or less. My understanding is that CTCF sites in *Drosophila* have a bimodal distribution and fall into two categories, proximal and distal to the promoter. It is possible that promoter sites are directly involved in transcription whereas distal sites, which tend to co-localize with SuHw sites, may have the enhancer blocking function.*

Response to the reviewer's comments

The reviewer asks whether *Drosophila* has a CTCFL homolog that also has the YDF motif. Revised **Supplementary Figure S2f** schematizes human and fly CTCF, cohesin subunits and cohesin regulators implicated in TAD boundary formation. Protein motifs reported to mediate direct protein-protein interactions are highlighted. This figure shows that humans have two CTCF proteins (CTCF and CTCFL) which differ in their ability to directly interact with cohesin via the YDF motif; and flies have a single CTCF protein.

This is a great suggestion. We classified CTCF peaks as TSS-proximal (TSS±200 bp) or TSS-distal (now described in the Methods): 15% of all CTCF peaks are TSS-proximal. (1) We measured **physical insulation score differences in *CTCF⁰* minus WT Hi-C maps, around TSS-proximal versus distal CTCF peaks.** (2) We counted the **number of differentially expressed (DE) genes with increased versus decreased expression in *CTCF⁰* relative to WT, around TSS-proximal CTCF peaks.** We find that:

1. TSS-distal CTCF peaks tend to form stronger physical boundaries than TSS-proximal CTCF peaks. This was visible in original Fig. 5d lane 6 and is now clearer in **new lane 7 classifying CTCF peaks associated with CTCF-bound contact domain boundaries as TSS-proximal or distal. Revised Fig. 5f shows that this trend is statistically significant.**
2. 17/108 (16%) TSS-proximal CTCF peaks have a DE gene within ±200 bp. Among these DE genes, 9 genes have increased and 8 have decreased expression in *CTCF⁰*, indicating that TSS-proximal CTCF does not have a specialized function in gene activation or silencing (though the small number of DE genes prevents a strong conclusion).
3. Also relevant to address the reviewer's comment - we highlight our original findings (Fig. 4c and Supplementary Fig. 4c) that TSS-proximal CTCF peaks (B, G and H) function as enhancer-blockers like TSS-distal CTCF peaks in a reporter assay. Enhancer-blocking function of CTCF peaks is also not limited to sites at which CTCF co-localizes with Su(Hw).